# Molecular correlates of cisplatin-based chemotherapy response in muscle invasive bladder cancer by integrated multi-omics analysis

Ann Taber [1,2,6], Emil Christensen [1,2,6], Philippe Lamy [1,6], Iver Nordentoft [1], Frederik Prip[1,2], Sia Viborg Lindskrog [1,2], Karin Birkenkamp-Demtröder [1,2], Trine Line Hauge Okholm [1,2], Michael Knudsen [1], Jakob Skou Pedersen [1,2], Torben Steiniche[3], Mads Agerbæk [4], Jørgen Bjerggaard Jensen[2,5] & Lars Dyrskjøt [1,2 ✉]

Overtreatment with cisplatin-based chemotherapy is a major issue in the management of muscle-invasive bladder cancer (MIBC), and currently none of the reported biomarkers for predicting response have been implemented in the clinic. Here we perform a comprehensive multi-omics analysis (genomics, transcriptomics, epigenomics and proteomics) of 300 MIBC patients treated with chemotherapy (neoadjuvant or first-line) to identify molecular changes associated with treatment response. DNA-based associations with response converge on genomic instability driven by a high number of chromosomal alterations, indels, signature 5 mutations and/or *BRCA2* mutations. Expression data identifies the basal/squamous gene expression subtype to be associated with poor response. Immune cell infiltration and high PD-1 protein expression are associated with treatment response. Through integration of genomic and transcriptomic data, we demonstrate patient stratification to groups of low and high likelihood of cisplatin-based response. This could pave the way for future patient selection following validation in prospective clinical trials.

[1] Department of Molecular Medicine, Aarhus University Hospital, Aarhus, Denmark. [2] Department of Clinical Medicine, Aarhus University, Aarhus, Denmark. [3] Department of Pathology, Aarhus University Hospital, Aarhus, Denmark. [4] Department of Oncology, Aarhus University Hospital, Aarhus, Denmark. [5] Department of Urology, Aarhus University Hospital, Aarhus, Denmark. [6] These authors contributed equally: Ann Taber, Emil Christensen, Philippe Lamy. ✉email: lars@clin.au.dk

Bladder cancer (BC) is the 9th most commonly diagnosed cancer worldwide and each year responsible for 165,000 deaths[1]. One of four patients with BC presents with muscle invasive disease. The standard of care for patients with localized muscle-invasive bladder cancer (MIBC) is neoadjuvant chemotherapy (NAC) followed by radical cystectomy. However, up to 40% of patients experience relapse after radical cystectomy and the vast majority of these succumb to the disease[2]. Cisplatin-based chemotherapy is recommended in both the neoadjuvant and first-line setting, with response rates up to ~50%[3,4]. However, no robust predictive biomarkers for chemotherapy response have entered into routine clinical use, and the inability to predict which patients will respond to chemotherapy represents a major clinical problem, as significant overtreatment of patients not responding is currently performed.

The genomic landscape of MIBC has been characterized as highly heterogeneous, with a high mutational burden and genomic instability caused by a multitude of processes like defects in DNA repair pathways, APOBEC induced mutagenesis, and overall high levels of mutagenesis from environmental chemical exposure[5–7]. Somatic mutations in DNA damage repair genes (DDR; e.g., *ERCC2*, *ATM*, *RB1*, and *FANCC*) have been correlated with cisplatin-sensitivity in MIBC[7–9], and the impact of *ERCC2* mutations has been demonstrated to drive cisplatin sensitivity in xenograft models[10]. However, *ERCC2* mutations have not been found to be associated with increased cisplatin sensitivity in all studies, highlighting that alternative and complex biological pathways may underlie treatment response mechanisms[11,12]. Ongoing clinical trials aim to validate DDR mutations as biomarkers for directing patients to bladder sparing approaches (e.g., NCT03609216). Transcriptional molecular classification of MIBC have shown the p53-like subtype to be correlated with poor response to NAC[13], and that patients with basal-like tumors showed increased survival following NAC compared to patients with basal tumors potentially treated with chemotherapy at the time of metastasis. Notably, the subtypes were, however, not shown to be significantly associated with pathologic response[14]. Furthermore, in a recent meta analysis, MIBC consensus subtypes were not significantly correlated with chemotherapy response[15]. Recently, we showed that circulating tumor DNA (ctDNA) measurements may constitute a powerful biomarker for monitoring treatment efficacy during NAC[12], indicating that a combined tumor centric and liquid biopsy approach may be a stronger tool for directing patients to optimal treatment.

Antitumor activities of chemotherapy besides DNA-synthesis and replication interference, have been shown to be promoted by the host immune system in several cancer types[16]. In BC it has been hypothesized that chemotherapy reinforces the antitumor immune response, and a higher ratio of cytotoxic T lymphocytes (CTLs) to activated regulatory T lymphocytes (Tregs) has been observed in responding patients[17]. However, the predictive value of the pre-treatment tumor microenvironment immune cell composition has received little attention.

Overall, predictive biomarkers of chemotherapy response have not shown consistency among studies - most likely due to small cohorts, inconsistency in treatment regimes, or treatment outcome reporting. Here, we report a multi-omics analysis of clinically well-annotated biological materials from 300 patients with BC, with the aim to identify molecular correlates of cisplatin-based chemotherapy response. We demonstrate that genomic instability driven by a high number of chromosomal alterations, indels, mutations in a tri-nucleotide signature 5 context and/or *BRCA2* mutations is associated with treatment response. Furthermore, we show that the basal/squamous gene expression subtype is associated with poor response and immune

cell infiltration and high PD-1 protein expression are associated with treatment response. Finally, through integration of genomic alterations and gene expression subtypes we identify patient groups with vastly different response rates. Our findings provide insight into the mechanisms associated with cisplatin-based treatment response, which could aid future personalized treatment in BC.

## Results

**Clinicopathological and multi-omics molecular data.** To investigate molecular correlates associated with treatment response, we included 300 tumors from patients with BC receiving chemotherapy; 62 received NAC before cystectomy (CX) and 245 received first-line chemotherapy upon detection of locally-advanced (T4b) or metastatic disease (Supplementary Fig. 1a; Supplementary Data 1; Supplementary Tables 1 and 2). An overview of the molecular analyses performed is provided in Supplementary Fig. 1b. Treatment response, defined as pathological noninvasive downstaging (≤pTa,CIS,N0) after NAC or complete or partial response after first-line treatment (RECIST 1.1), was observed for ~57% of patients (n = 172/300, NAC: ~63%, first-line: ~55%).

**Tumor-specific DNA alterations.** WES was performed using DNA from 165 tumors (76× median coverage, median of 92% target bases at 20X) and associated leukocyte germline DNA (46× median coverage, median of 85% target bases at 20X). A median of 524 (35–8231) mutations and 41 (6–2171) indels per tumor were identified. An overview of genomic data from WES is presented in Fig. 1a and Supplementary Fig. 2a (focus on significantly mutated genes in this study and TCGA cohort, respectively). For patients responding to chemotherapy, we observed a significantly higher number of indels (p = 0.031; SNVs: p = 0.38; neoantigens: p = 0.17; Wilcoxon rank-sum test; Fig. 1b–d) and a significantly higher proportion of the genome under allelic imbalance (SNP arrays (n = 49); p = 0.024; Wilcoxon rank-sum test; Fig. 1e), indicating that a more disrupted genome is more sensitive to treatment with chemotherapy. To further address genome disruption, we computed the microsatellite instability (MSI) status for all patients, however, MSI status was not associated with genome disruption or chemotherapy response in this study (Supplementary Fig. 2b). In addition, we identified chromosome 18 and regions 2p25.3-p22.1, 11p15.5-p11.2, and 12q13.3-q21.31 to be more affected by allelic imbalance in patients responding to chemotherapy compared to patients not responding (Supplementary Fig. 3). We, furthermore, investigated nonsense and missense mutations classified as damaging (according to PolyPhen-2[18] and MutationAssessor[19]) in DDR related genes (Supplementary Table 3), but found no association between DDR gene mutation status and chemotherapy response (Fig. 1f).

Following, we conducted analysis of mutational processes using the previously identified signatures in BC[5,6]: Single Base Substitution 1(SBS1;age-related), SBS2 and SBS13 (APOBEC related), and SBS5 (*ERCC2* mutation related[20]). Samples whose mutational trinucleotide profile could not be well explained by only using these four signatures were not considered for signature analysis. Overall, tumors could be assigned to two major groups based on the mutational process responsible for the majority of the mutations: SBS2+13 associated tumors (SBS APO) and SBS5 associated tumors (Fig. 1a). SBS APO tumors showed a higher number of SNVs (p = 3.1e−7; Wilcoxon rank-sum test; Fig. 1g) and SBS5 tumors harbored more indels (p = 0.054; Wilcoxon rank-sum test; Fig. 1h). Proportion of the genome under allelic imbalance did not differ between the two groups

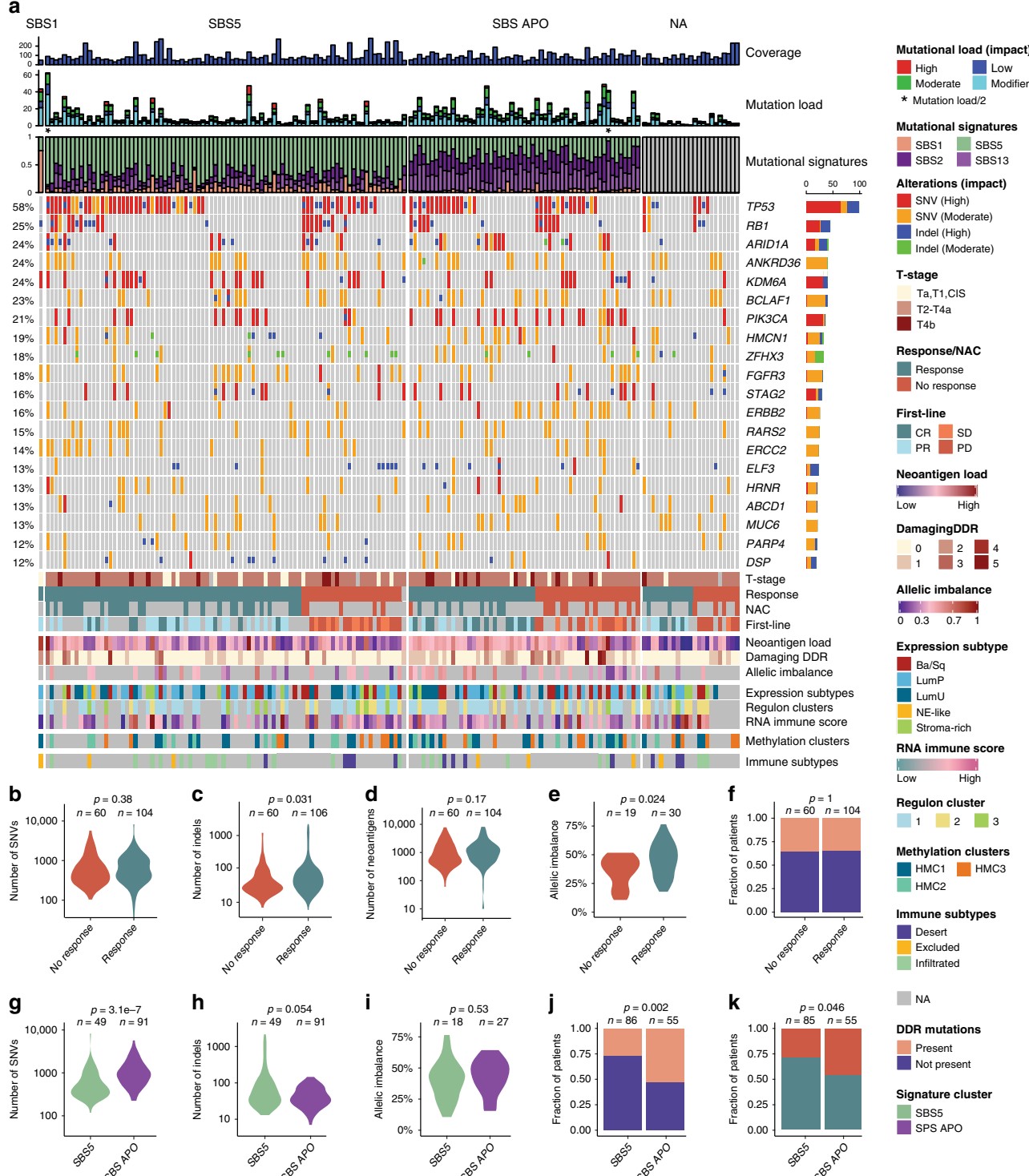

**Fig. 1 Overview of the genomic alterations correlated to chemotherapy response. a** Oncoplot showing the significantly mutated genes in 165 tumors annotated by exome coverage, mutation load stratified by impact (as defined by SnpEff) and mutational signature deconvolution (top panels) and by clinical response, neoantigen load, number of damaging mutations in DDR genes, percentage of genome in allelic imbalance, expression subtypes, regulon cluster, RNA immune score, hypermethylation cluster and immune phenotype (bottom panel). **b–e** Violin plots showing the total number of SNVs, indels, neoantigens or the percentage of the genome in allelic imbalance compared to chemotherapy response. **f** Presence of damaging DDR mutations compared to chemotherapy response. **g–i** Violin plots showing the total number of SNVs, indels or the percentage of the genome in allelic imbalance compared to the two main mutational subtypes. **j** Presence of DDR mutations compared to the two main mutational subtypes. **k** Mutational subtypes compared to chemotherapy response. All *p*-values were calculated using a Wilcoxon rank-sum test for continuous variables and a Fisher's exact test for categorical variables. Source data are provided as a Source data file.

(Fig. 1i). However, we observed that SBS APO tumors contained a significantly higher fraction of tumors with damaging DDR gene mutations compared to SBS5 tumors ($p = 0.002$; Fisher's exact test; Fig. 1j). Interestingly, patients with SBS5 tumors showed a higher response rate to chemotherapy compared to patients with SBS APO tumors ($p = 0.046$; Fisher's exact test; Fig. 1k).

We further characterized the mutational landscape by investigating the number of mutations and specific genes associated with each major mutational signature. We identified a higher number of mutations in a SBS5 context among patients responding to chemotherapy (Fig. 2a). This was not observed for SBS2+13 or SBS1 (Supplementary Fig. 4a). *ERCC2* mutated tumors have previously been associated with a high number of SBS5 mutations[20]. Here, we observed no association between *ERCC2* mutated tumors and elevated response rates (Fig. 2b), as previously observed[8]. We consequently hypothesized that other genes may be associated with a high number of SBS5 mutations and an elevated response rate. We delineated somatic mutations (high and moderate impact defined by SnpEff[21]) associated with a SBS5 dominated mutational landscape and identified *BRCA2* and *ERCC2* mutated tumors to be significantly enriched for SBS5 mutations ($p = 0.0003$, $p = 0.00003$, respectively; permutation test; Fig. 2c), together with a few other genes like *ABCA6*, *TEP1*, and *BOD1L1* ($p = 0.00078$, $p = 0.00085$, $p = 0.00168$, respectively; permutation test; Fig. 2c). *BRCA2* and *ERCC2* mutated tumors were not significantly enriched for mutations in an APOBEC (SBS2+13) context (Fig. 2d). Importantly, patients with *BRCA2* mutated tumors had a significantly higher response rate to chemotherapy compared to patients with *BRCA2* wild-type (wt) tumors ($p = 0.017$; Fisher's exact test; Fig. 2e). Kaplan–Meier survival analysis indicated improved survival for *BRCA2* mutated patients ($p = 0.11$; log-rank test; Fig. 2f). In TCGA data, *BRCA2* and *ERCC2* mutated samples similarly demonstrated elevated numbers of SBS5 mutations, however, they also demonstrated elevated numbers of mutations in an APOBEC (SBS2 +13) context (Fig. 2g, h and Supplementary Fig. 4b). Importantly, in TCGA, all patients with *BRCA2* mutated tumors receiving cisplatin-based chemotherapy ($n = 6$ out of 62 patients) responded (vs. 34/56 for *BRCA2* wt, $p = 0.08$; Fisher's exact test; Fig. 2i), while 6/7 patients with *ERCC2* mutated tumors responded (vs. 34/55 for *ERCC2* wt, $p = 0.40$; Fisher's exact test; Fig. 2j).

**Genomic analysis of paired primary and metastatic lesions.** In this study, we assessed tumor-specific alterations associated with chemotherapy response based on primary tumor specimens, however in ~30% (49/165 patients) the primary tumors were removed, and chemotherapy response evaluation was based on image analysis of the metastatic sites. To assess if molecular correlates of chemotherapy response identified in primary tumors persist in metastatic lesions, we performed WES of DNA from available metastatic lesions ($n = 11$) from six patients (Fig. 3a) to a median read depth of 100X. The clonal relationships between primary tumors and associated metastatic lesions are illustrated in Fig. 3b. We identified a median of 390 mutations shared between primary tumors and metastatic lesions (range: 173–1350) and a median of 29 mutations (range: 0–87) in primary tumors that were not present in the respective metastatic samples. This corresponds to a median fraction of only 7% (range: 0–23%) of all mutations in primary tumors that were not carried over to metastatic lesions (Fig. 3c). In metastatic samples, the mutations shared with primary tumors demonstrated significantly higher allele frequencies compared to private mutations (Fig. 3d). This suggests that most of the primary tumor mutations appeared to be clonal in the metastasis. Additionally, the large

number of novel low-frequency mutations in metastatic lesions probably indicates the simultaneous development of multiple subclones.

Our primary tumor analysis demonstrated a higher chemotherapy response rate for patients with tumor mutational landscapes dominated by SBS5. We, therefore, compared the mutational processes underlying the mutational landscape of the primary tumor to that of the metastases (Fig. 3e). In three patients, the majority of both shared and private mutations were observed in an SBS5 context. For the remaining three patients, the majority of trunk mutations were observed in a SBS2+13 (APOBEC) context while the majority of private mutations in the metastatic samples were observed in an SBS5 context. This suggests APOBEC mutagenesis may primarily occur before metastatic dissemination, whereas SBS5 shapes the mutational landscape of metastatic lesions.

**Gene expression analysis.** Gene expression consensus subtypes were called based on RNA-Seq data ($n = 121$)[15] with the following distribution: LumP: 36.4%, LumNS: 0%, LumU: 21.5%, Stroma-rich: 14.9%, Ba/Sq: 26.4%, NE-like: 0.8%. Assigned subtypes were validated based on expression of basal-, luminal-, immune-, extracellular matrix- and *FGFR3*- related markers (Fig. 4a). We observed a high response rate in stroma-rich tumors and a low response rate in Ba/Sq tumors, but the overall difference in response rates was not significant ($p = 0.11$; Fisher's exact test), as previously reported[12,15]. However, we did observe a significantly lower response rate in Ba/Sq tumors compared to the other tumor subtypes ($p = 0.030$; Fisher's exact test; Fig. 4b). Overall survival was significantly reduced in patients with Ba/Sq tumors compared to patients with other gene expression subtypes ($p = 0.034$; log-rank test; Fig. 4c). Integration of WES data revealed similar numbers of mutations between gene expression subtypes, however, LumU tumors harbored significantly more mutations than LumP tumors ($p = 0.007$; Wilcoxon rank-sum test; Fig. 4d), as expected[15]. Similar numbers of indels were observed between the gene expression subtypes ($p = 0.65$; Kruskal–Wallis rank-sum test). We observed no associations between gene expression subtypes and clusters based on mutational signatures ($p = 0.86$; Fisher's Exact test).

We further explored the transcriptional landscape by analyzing gene expression of a group of co-regulated genes associated with a predefined list of transcription factors (i.e., regulons; Supplementary Table 4). Three clusters were identified showing distinct regulon activity patterns (Fig. 4e). Regulon clusters were found to overlap with expression subtypes: 89% of the tumors in regulon cluster 1 (R1) were luminal, 100% of the tumors in regulon cluster 2 (R2) were basal or stroma-rich, and regulon cluster 3 (R3) contained a mixture of expression subtypes (Fig. 4f). No significant association between regulon clusters and chemotherapy response was observed (Fig. 4g).

Previous studies have identified an association between immune cell infiltration and response to chemotherapy, suggesting that the presence of immune cells increases treatment efficacy[16,22]. We, therefore, quantified the presence of immune cells in the tumors by deconvolution of the bulk RNA-Seq data (Fig. 4e). We observed significant differences between expression subtypes and immune cell infiltration, i.e., basal and stroma-rich tumors showed a higher level of immune infiltration compared to luminal tumors (Fig. 4h). Immune infiltration per se was, however, not associated with response to chemotherapy (Fig. 4i and Supplementary Fig. 5a). Grouping patients based on immune infiltration and neoantigen load neither displayed an association with response to chemotherapy (Supplementary Fig. 5b). We,

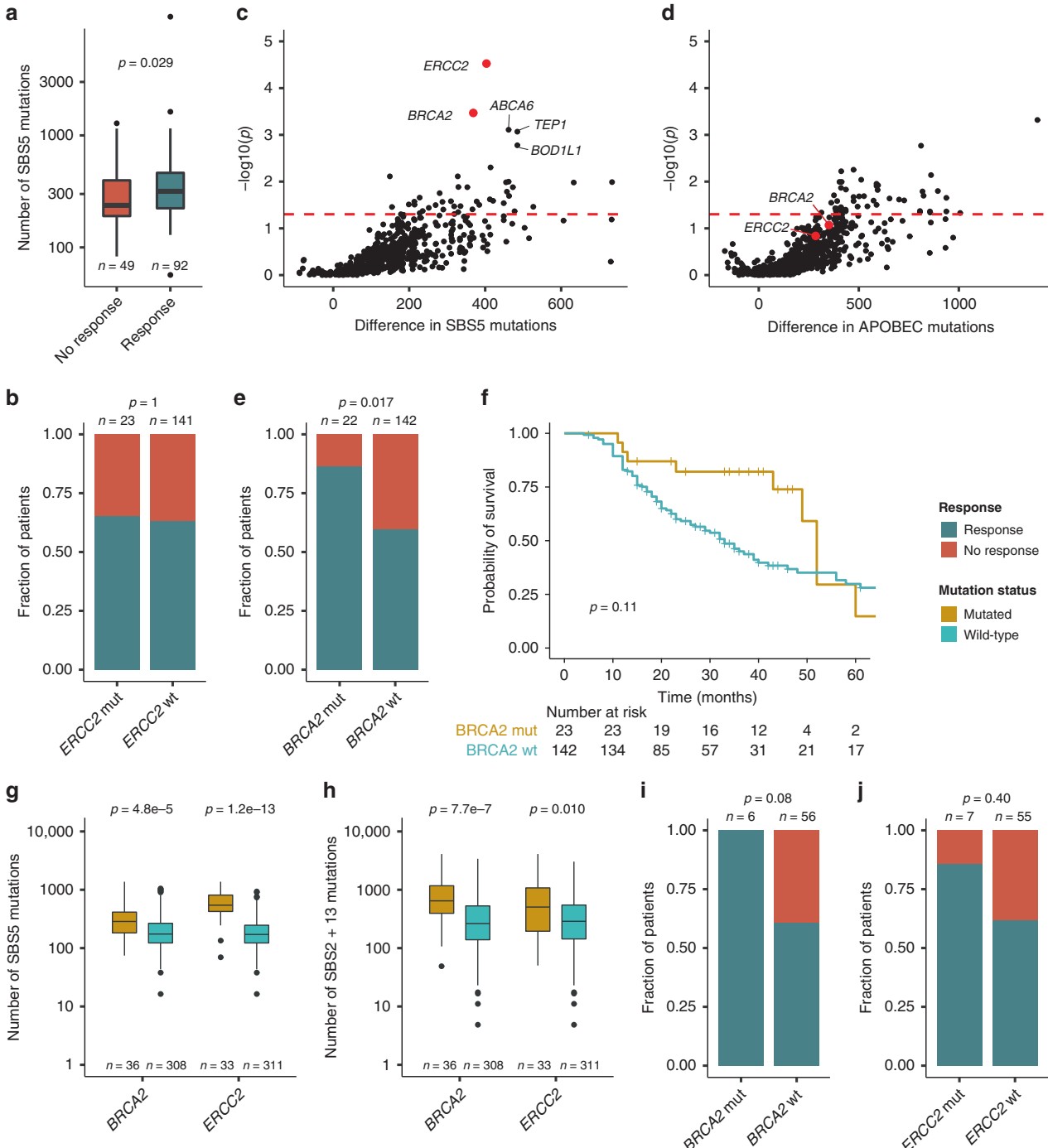

**Fig. 2 _BRCA2_ mutated tumors are associated with mutations in an SBS5 context and response to chemotherapy. a** Number of mutations in an SBS5 context in relation to response to chemotherapy. **b** Mutation status of _ERCC2_ compared to chemotherapy response. **c, d** Volcano plots showing the difference between the median number of mutations for mutated tumors and the median number of mutations for wild-type tumors for all genes mutated in more than 5% of the patient cohort (only mutations with moderate- or high protein impact are considered). The red dashed lines indicate significance levels at $p = 0.05$. **c** Represents the number of mutations in an SBS5 context and **d** represents the number of mutations in an SBS2+13 (APOBEC) context. _P_-values were calculated using a permutation test ($n = 100,000$) that controls for mutation burden per sample and gene. **e** Mutation status of _BRCA2_ compared to chemotherapy response. **f** Kaplan–Meier survival analysis depicting the probability of overall survival stratified by _BRCA2_ mutation status. **g–j** Based on TCGA data. **g** Number of mutations in an SBS5 context for _ERCC2_ and _BRCA2_ wild-type and mutant samples in TCGA data. **h** Number of mutations in an SBS2+13 (APOBEC) context for _ERCC2_ and _BRCA2_ wild-type and mutant samples in TCGA data. **i, j** Mutation status of _ERCC2_ and _BRCA2_ compared to response to chemotherapy in TCGA data. _P_-values were calculated using a Fisher's exact test for categorical variables, a Wilcoxon rank-sum test for continuous variables and a log-rank test for comparing survival curves. For all boxplots, the center line represents the median, box hinges represent first and third quartiles, whiskers represent ±1.5 × interquartile range (IQR) and points represent outliers. Source data are provided as a Source data file.

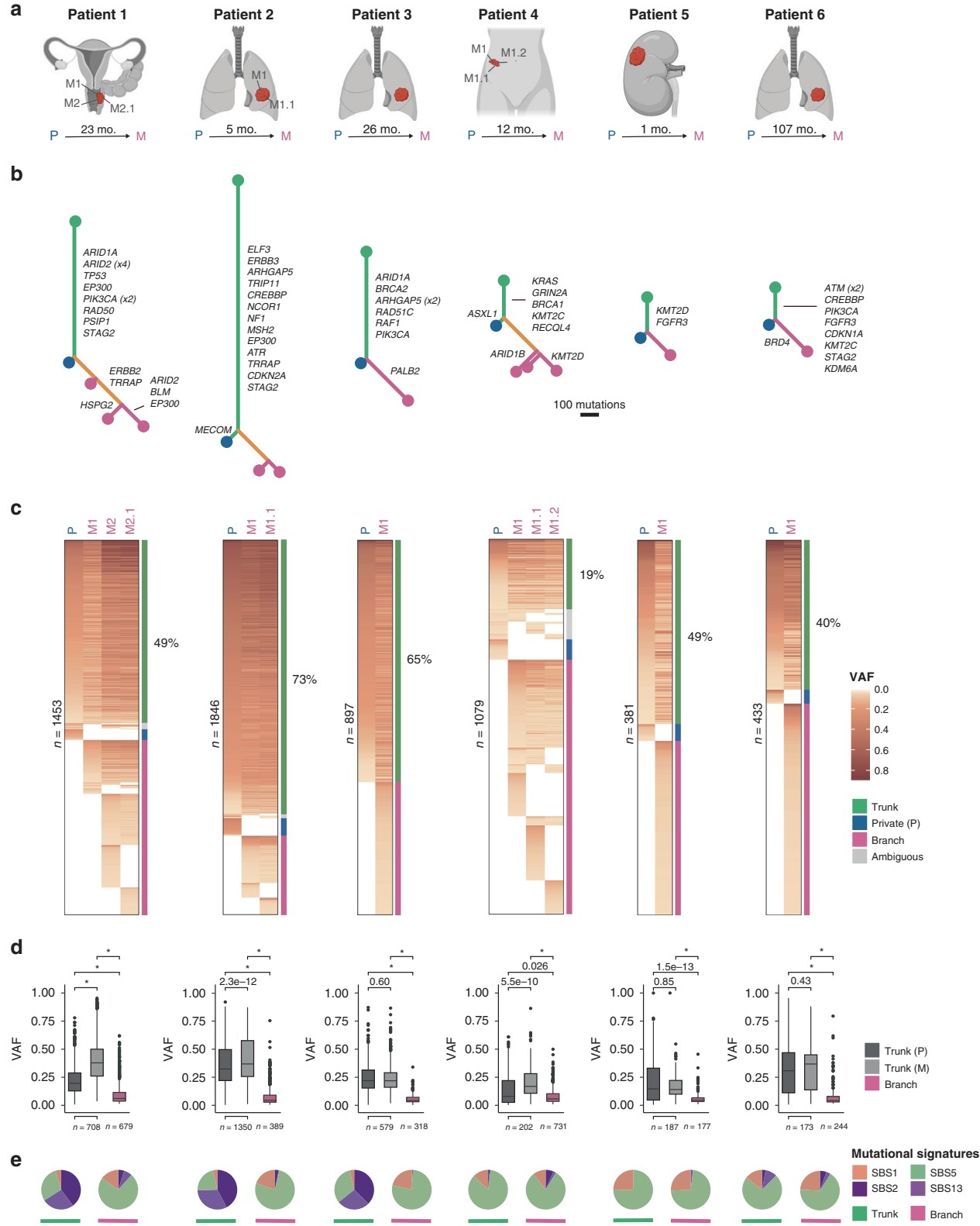

therefore, speculated whether the presence of mutations and an effective machinery for neoantigen presentation would be a prerequisite for an effective immune response. Overall, the number of mutations and indels showed no association with immune infiltration levels (Supplementary Fig. 5c). However, for tumors with high levels of immune infiltration, we found a trend towards higher expression of neoantigen presenting genes[23] for

patients responding to chemotherapy compared to non-responders ($p = 0.069$; Wilcoxon rank-sum test; Fig. 4j).

**Methylation alterations**. Illumina EPIC arrays were used to investigate if methylation alterations were associated with chemotherapy response ($n = 72$). We performed unsupervised

**Fig. 3 Delineation of metastatic evolution before chemotherapy.** All six patients were treated with cisplatin-based chemotherapy upon detection of metastatic disease. **a** Location of metastasis and the time to recurrence (mo = months). P = primary tumor, M = metastasis. The images were created using Biorender.com. **b** Clonal relationships between primary tumor samples and metastatic lesions depicted by phylogenetic trees. Trunk/branch lengths correspond to the number of SNVs. Mutations in genes involved in DDR, frequently mutated in TCGA or identified as drivers in BC (IntOGen[62]) are indicated. Green = trunk, yellow = branch, blue = primary tumors, pink = metastatic lesions. **c** Variant allele frequencies for mutations identified in either of the available samples per patient. Identified mutations were subjected to read-counting in processed bam files to enable identification of mutations called in one sample, and present, but not called in another sample. The required read depth for identifying a given mutation was calculated for every position based on the lowest observed allele frequency. Only positions with sufficient read depth in all investigated samples were included. **d** Box plots depicting the observed allele frequencies for trunk and branch mutations. Asterisks indicate p-values below 2.2e−16. **e** Relative contribution of mutational signatures in the trunks (left circles) and branches (right circles). P-values were calculated using a Wilcoxon rank-sum test. For all boxplots, the center line represents the median, box hinges represent first and third quartiles, whiskers represent ±1.5 × interquartile range (IQR) and points represent outliers. Source data are provided as a Source data file.

consensus clustering using tumor-associated hyper- or hypo-methylated CpG sites (Fig. 5a and Supplementary Fig. 6). For hypermethylated CpG sites, we identified three major hypermethylation clusters (HMC1–3): two extreme clusters, HMC2 and HMC3, and one intermediate, HMC1 (Fig. 5a). HMC2 almost exclusively consisted of samples with luminal gene expression subtypes (83% luminal vs 61% and 31% for HMC1 and 3, respectively; Fig. 5b) and showed a methylation profile very dissimilar to normal bladder tissue and leukocytes. Contrarily, HMC3 was highly similar to normal bladder tissue and leukocytes and showed a higher fraction of samples with stroma-rich gene expression subtypes (37% stroma-rich vs 6% and 0% for HMC1 and 2, respectively; Fig. 5b). In addition, HMC3 showed significantly higher gene expression scores attributable to immune infiltration, stroma content and microenvironment contribution. Similar gene expression scores for smooth muscle cells were observed between methylation clusters (Fig. 5c). Collectively, these findings indicate that HMC3 represents a group of tumors characterized by high immune cell infiltration and probably few carcinoma cells. We observed no correlation between methylation clusters and response to chemotherapy (Fig. 5d).

**Spatial proteomics analysis by digital pathology.** We performed multiplex immunofluorescence and immunohistochemical staining to analyze the spatial composition of tumor- and immune cells (T-Helper, CTLs, Tregs, B-cells, M1, and M2 macrophages) and immune evasion mechanisms (PD-L1, PD-1, and MHC class I) in 184 patients who received first-line chemotherapy. All antibodies used are listed in Supplementary Data 2. Automated image analysis algorithms were developed to compare fractions of selected markers intratumoral versus peritumoral (Supplementary Fig. 7). Based on the spatial tumor- and immune cell compositions we grouped the tumors into three main immune subtypes: immune infiltrated, immune excluded and immune desert, as previously reported[24,25] (Fig. 6a, b). Response to chemotherapy was more frequently observed in patients with the immune excluded and infiltrated subtypes compared to the immune desert subtype (p = 0.006; chi-square test; Fig. 6c). No difference in overall survival was observed between immune subtypes (p = 0.12; log-rank test; Fig. 6d). In addition, we investigated immune evasion mechanisms and found that a high fraction of intratumoral or peritumoral PD-1 positive cells was significantly associated with treatment response (p = 0.008, p = 0.002, respectively; Wilcoxon rank-sum test; Fig. 6e). No correlation between PD-L1 positive cells or MHC class I on tumor cells and treatment response was observed (Fig. 6f, g). A combined analysis of intratumoral PD-1 and PD-L1 expression also showed significant correlation with chemotherapy response (p = 0.030; chi-square test; Fig. 6h). We further explored immune evasion mechanisms within immune subtypes and found that immune infiltrated tumors were accompanied by a high frequency of cells displaying intratumoral PD-1, PD-L1, and MHC expression (Fig. 6i–k). These features suggest that immune cells are able to migrate into the tumor parenchyma, however, the antitumor response might be less effective due to immunosuppression. Immune desert tumors showed a lower fraction of PD-1 positive cells compared to the other subtypes (Fig. 6i). Furthermore, downregulation of MHC class I on tumor cells was observed in the immune desert and excluded tumors, which could explain the lack or retention of immune cells in the stroma surrounding the tumor parenchyma (Fig. 6k).

**Integrative multi-omics analysis.** An integrative analysis of genomics, transcriptomics and proteomics data enabled us to further examine possible biological differences between responders and non-responders of chemotherapy. The genomic features associated with response in this study seemed to converge on inefficient DNA damage response. We, therefore, assigned patients to high and low genomic instability groups based on SBS5 mutations, indels, allelic imbalance and BRCA2 mutation status. Patients with high genomic instability had a response rate of 71% vs. 49% for patients with low genomic instability (p = 0.007; chi-square test; Fig. 7a). For integration with transcriptomic and proteomics data, we focused on patients with gene expression subtypes assigned using RNA-seq data (n = 121; Fig. 7b). The Ba/Sq gene expression subtype was associated with a lower response rate compared to patients with other subtypes (p = 0.027; chi-square test; Fig. 7c). For further investigation of the predictive value of molecular and clinical features, we dichotomized the data for every variable and calculated odds ratios for chemotherapy response. Indels, SBS5 mutations, BRCA2 mutations, level of genomic instability, Ba/Sq gene expression subtype, and IHC-based immune subtypes showed a significant association with response (p = 0.004–0.021; Fisher's exact test; Fig. 7d). Molecular variables were generally more significant for patients whose response was evaluated based on the primary tumors (NAC) compared to metastatic lesions (first line), but similar trends were observed. Among the clinical features, only performance status prior to first-line chemotherapy displayed a significant association with response (p = 0.025; Fisher's exact test; Fig. 7d). Since the majority of patients (~98%) were treated with cisplatin-based chemotherapy, we integrated genomic instability measures and gene expression subtypes for these patients to assess the impact on response rates. This revealed that while high genomic instability is associated with elevated response rates, the Ba/Sq subtype was associated with reduced response rates across the genomic instability groups (Fig. 7e). Importantly, this combined analysis identifies a group of patients with a very high response rate (80%; NAC: 90%; First-line: 71%) characterized by high genomic instability and non-Ba/Sq gene expression subtype and a group of patients with a very low response rate (25%; NAC: 20%; First-line: 29%) characterized by low genomic instability

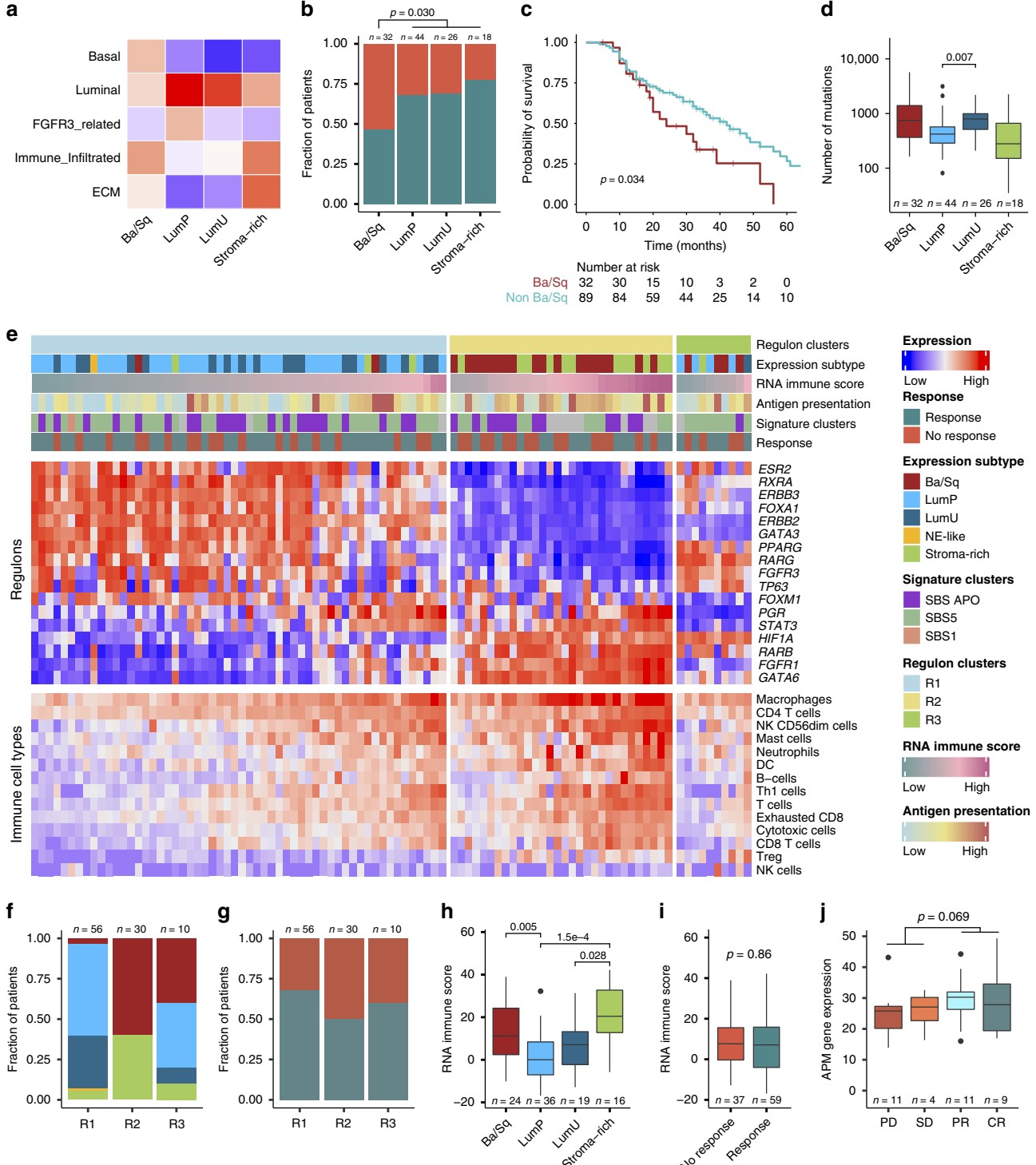

**Fig. 4 Gene expression characteristics and relation to chemotherapy response. a** Visualisation of four identified consensus gene expression subtypes by selected gene sets. **b** Gene expression subtypes compared to chemotherapy response. **c** Kaplan–Meier survival analysis showing the probability of overall survival for patients with and without Ba/Sq gene expression subtype. **d** Number of mutations for gene expression subtypes. **e** Heatmap showing relative expression values for identified regulons and deconvoluted immune cells. **f** Regulon clusters compared to gene expression subtypes. **g** Regulon clusters in relation to response to chemotherapy. **h** Immune score across the identified gene expression subtypes. **i** Immune score compared to response to chemotherapy. **j** Summarized expression of the antigen-presenting machinery for immune hot (above median immune score) samples stratified by RECIST 1.1 response values. Only one tumor was classified as NE-like and was therefore omitted from this figure. Missing data is depicted in gray. P-values were calculated using a Fisher's exact test for categorical variables, a Wilcoxon rank-sum test for continuous variables and a log-rank test for comparing survival curves. For all boxplots, the center line represents the median, box hinges represent first and third quartiles, whiskers represent ±1.5 × interquartile range (IQR) and points represent outliers. Source data are provided as a Source data file.

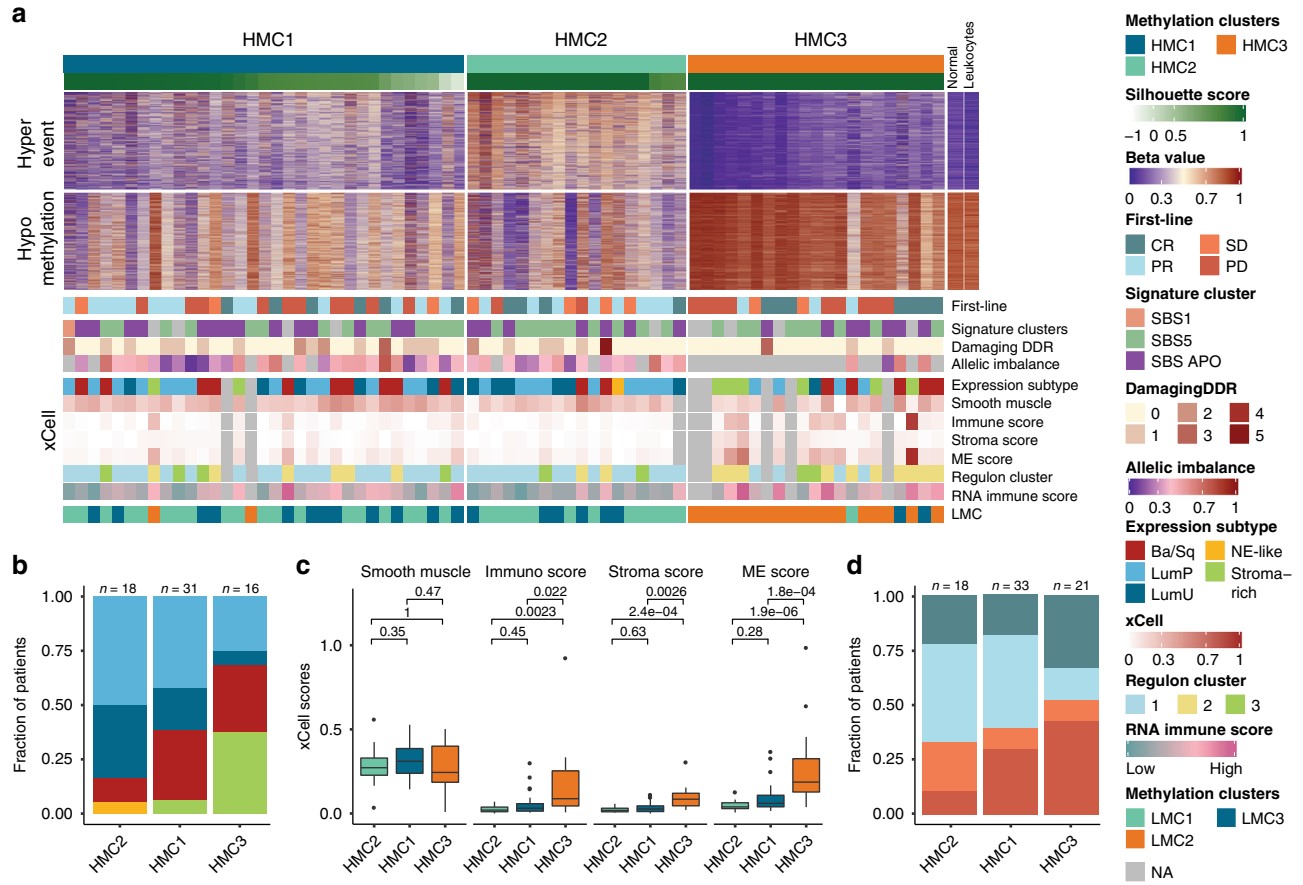

**Fig. 5 DNA methylation subtypes based on hypermethylated cancer-specific CpG sites. a** Clustering of samples based on hypermethylation events (*n* = 5000). Heatmap shows beta values and the right panel shows normal bladder and leukocyte beta values for comparison. **b** Methylation clusters compared to gene expression subtypes. **c** Gene set scores calculated using xCell and stratified by methylation clusters (HMC2: *n* = 17; HMC1: *n* = 31; HMC3: *n* = 16). **d** RECIST 1.1 response measurements stratified by methylation clusters. MEscore = Microenvironment score. *P*-values were calculated using a Wilcoxon rank-sum test. For all boxplots, the center line represents the median, box hinges represent first and third quartiles, whiskers represent ±1.5 × interquartile range (IQR) and points represent outliers. Source data are provided as a Source data file.

and Ba/Sq gene expression subtype (*p* = 4.3e-4; NAC: *p* = 0.006; First-line: *p* = 0.077; chi-square test; Supplementary Fig. 8).

## Discussion

To capture the complex interplay between tumor biology and possible cisplatin-based chemotherapy response and resistance mechanisms, we investigated genomics, transcriptomics, epigenetics, and proteomics with clinical and pathological data. The predictors of chemotherapy response in this patient cohort were: a large proportion of the genome under allelic imbalance, a high number of indels, a high number of SBS5 mutations, somatic mutations in *BRCA2*. Furthermore, the Ba/Sq gene expression subtype seemed to suppress chemotherapy efficacy. In addition, we observed that immune infiltrated and excluded subtypes, and elevated PD-1 protein expression were associated with chemotherapy response. Consequently, chemotherapy response is associated with a multitude of parameters.

Cisplatin exerts its cytotoxic properties by formation of DNA cross-links, which inhibit DNA replication and transcription, consequently promoting cell death. DDR mechanisms play an important role in the repair of the cytotoxic DNA damage[26]. *ERCC2*, a key component in the nucleotide excision repair complex, has been associated with improved response to cisplatin-based chemotherapy in MIBC[10]. Similarly, mutations in other DDR related genes have been associated with response to chemotherapy in MIBC[9,11]. Here, we found mutations in the

homologous recombination associated gene *BRCA2* to be associated with elevated chemotherapy response rate. This is in line with findings from ovarian cancer[27]. A recent study reported that germline *BRCA2* and other DDR gene mutations were frequently observed in patients with BC[28]. Clinical trials investigating the use of PARP inhibitors in patients with metastatic BC and somatic or germline *BRCA2* or other DDR mutations are currently ongoing (NCT03397394, NCT03448718, and NCT03375307). In this study, *BRCA2* mutated tumors were, similar to *ERCC2* mutated tumors, associated with high numbers of mutations in a SBS5 context, indicating that this mutational signature is an indicator of a DDR-deficient tumor that may be more sensitive to chemotherapy. This is also in line with our findings that *ERCC2* mutations *per se* were not associated with treatment response in this cohort - but the SBS5 mutational signature was.

We analyzed primary tumor samples from patients receiving both immediate NAC and later first-line treatment for metastatic disease. Previous studies have identified extensive temporal heterogeneity between primary tumors and metastatic lesions[29,30]. Here we demonstrated that the majority of mutations in the primary tumors were conserved and clonal in the metastatic lesions. However, we also observed a high number of novel mutations at low allele frequencies in the metastatic lesions. These observations could reflect overall high similarity to the primary tumors but with concurrent development of multiple novel

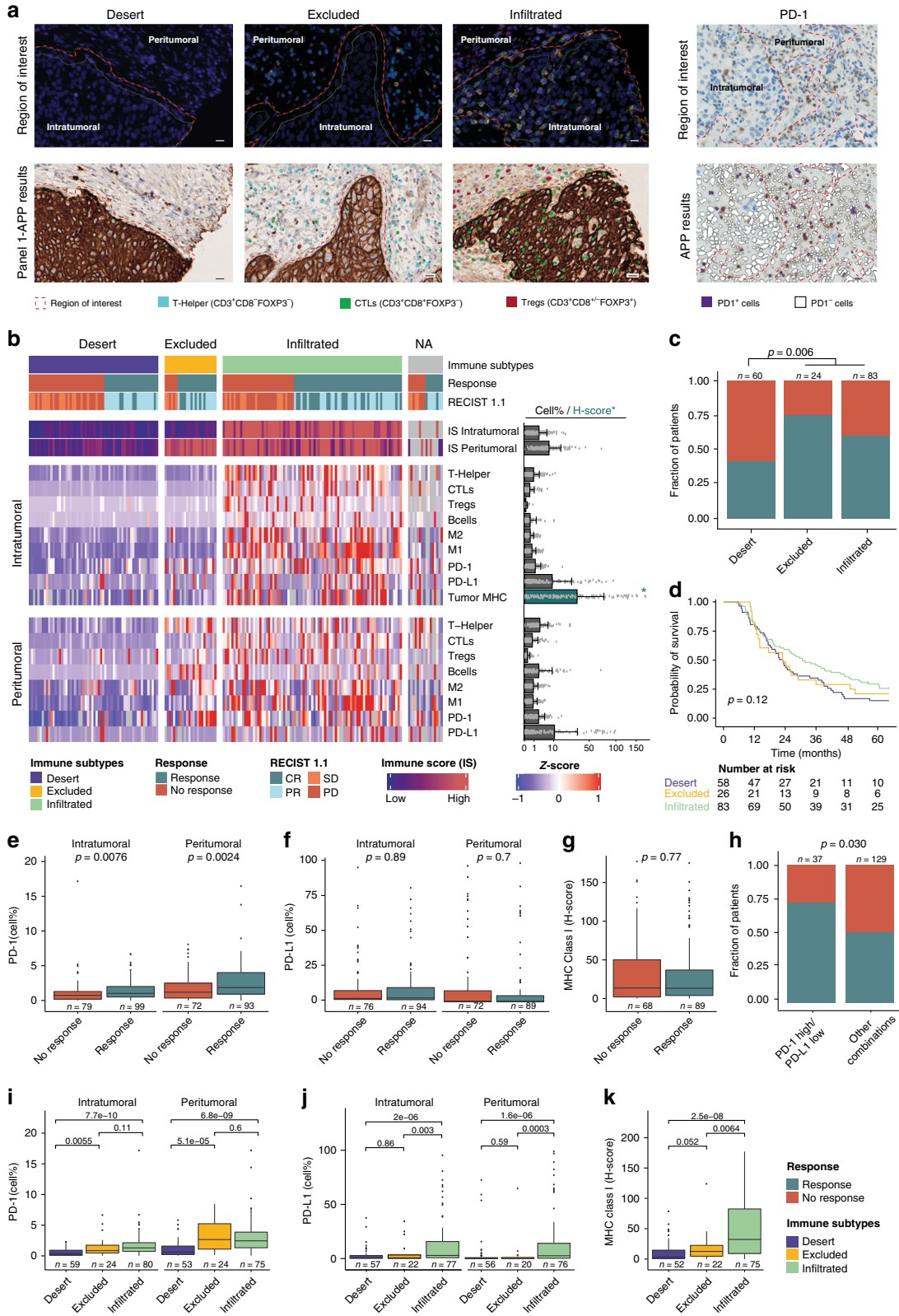

subclones. This observation might be impacted by the fact that DNA from primary tumors was extracted from frozen tissue while DNA from metastatic lesions was extracted from formalin-fixed paraffin embedded (FFPE) tissue. The usage of different WES library pipelines might also have impacted these observations, however our analysis only considers genomic positions with sufficient data across all samples in order to minimize the impact of potential systematic differences.

Previous studies comparing gene expression subtypes to chemotherapy response have shown conflicting results. Choi et al. found similar response rates for basal and luminal tumors and a lower response rate for p53-like tumors[13]. Seiler et al. showed

**Fig. 6 Immune tumor microenvironment analysis by spatial proteomics.** Immunostaining performed on bladder cancer tissue microarrays from 184 patients. All protein measurements were performed once for each distinct sample. **a** Staining results shown for multiplex immunofluorescence (mIF) panel 1 and PD-1 with corresponding image analysis application (APP) of four tumors representing each immune subtype and high PD-1 expression, respectively. Red dashed lines divide tissue into intratumoral and peritumoral regions of interest. Scale bar: 20 μm. **b** Spatial organisation of immune cells and immune evasion mechanisms stratified by immune subtypes. Heatmap shows z-scores and the barplot the mean + SD immune scores (IS), cell percentages, or H-score [1 × (% cells low intensity) + 2 × (% cells moderate intensity) + 3 × (% cells high intensity)] for MHC-expression on carcinoma cells. Asterisks denote the barplot representing the H-score. Points represent the corresponding data points. **c** Immune subtypes compared to chemotherapy response. **d** Kaplan–Meier survival curves showing overall survival (OS) stratified by immune subtypes. **e–g** Intratumoral and peritumoral fractions of immune evasion mechanisms (PD-1, PD-L1, MHC) compared to chemotherapy response. **h** Intratumoral combined PD-1/PD-L1 expression stratified by chemotherapy response. PD-1 high/PD-L1 low was compared to PD-1 high/PD-L1 high, PD-1 low/ PD-L1 low and PD-1 low/PD-L1 high combined. **i–k** Relationship between immune evasion mechanisms and immune subtypes. Statistical significance was assessed using a chi-square test for categorical variables, a Wilcoxon rank-sum test for continuous variables and a log-rank test for comparing survival curves. For all boxplots, the center line represents the median, box hinges represent first and third quartiles, whiskers represent ±1.5 × interquartile range (IQR) and points represent outliers. Source data are provided as a Source data file.

increased survival for patients with basal tumors in a NAC treated cohort compared to the TCGA cohort[14]. A recent meta analysis found no significant difference in response rates between subtypes[15]. However, in this study, we observed a lower response rate for patients with basal tumors. The conflicting observations could indicate that chemotherapy response is governed by a multitude of factors and that integration with other data types might be necessary to understand the biology behind response.

Platinum-based chemotherapy is known to induce immunogenic cell death, which ultimately improves anticancer effects of cisplatin through increased antigen presentation and the following T cell response[16,22]. In the present study, we showed that a high fraction of intratumoral and peritumoral PD-1 positive cells correlated with treatment response, whereas no association between PD-L1 and response was observed. High PD-1 expression, associated with activated T-cells, B-cells, macrophages, and NK-cells[31], may indicate the presence of an activated pretreatment antitumor immune response that further enhances treatment efficiency. This theory is supported by our findings that immune infiltrated and excluded subtypes had a better treatment outcome - in fact, tumors with a high peritumoral PD-1 expression showed the best response rates. Consequently, these patients may benefit from a combination of immunotherapy and chemotherapy, however additional studies are needed to fully establish the predictive value of PD-1 expression in association to chemotherapy response.

This study is limited by partially overlapping multi-omics analyses. Including both patients receiving chemotherapy in the neoadjuvant setting (i.e. evaluated by pathologic response in the primary tumor) and patients with advanced disease (i.e., evaluated by RECIST 1.1) represent another inherent limitation of the present study. Although pathological downstaging has been used for evaluating NAC response in several studies[3,32], it is not without limitation, including the potential impact of previous TURB-T on the rate of downstaging. Overall, our results highlight several molecular correlates of chemotherapy response and importantly, the integration of genomic instability and gene expression subtypes identified patient groups with vastly different response rates. If successfully validated in future prospective trials, these findings could aid in selecting patients with a high probability of treatment response and potentially minimize the current overtreatment of patients. Prospective validation is currently ongoing (NCT04138628).

## Methods

**Patient details**. A total of 300 patients with BC receiving chemotherapy were included in the study; 62 received NAC before cystectomy (CX) and 245 received first-line chemotherapy upon detection of locally-advanced (T4b) or metastatic disease (Supplementary Fig. 1a). Treatments were carried out according to Danish National guidelines, which adhere to the European Guidelines for BC[33]. Cisplatin-

based chemotherapy was administered in ~98% of cases, however, for the purpose of this study no distinction was made between the different chemotherapy regimens described in Supplementary Data 1. Pretreatment staging was based on cross-sectional imaging (baseline) and pathological assessment of TURB-T (transurethral resection of bladder tumor) specimen. NAC treatment response was defined as pathological non-invasive downstaging (≤pTa,CIS,N0) based on examination of the CX specimen. First-line treatment response was defined as complete (CR) or partial response (PR) based on post-treatment cross-sectional imaging according to the RECIST 1.1 guidelines (Response Evaluation Criteria in Solid Tumors)[34]. 55.5% (136/245 patients) had an intact bladder at the time of first-line treatment. Post-treatment pathological staging of the residual tumor was evaluated in patients with radiologic CR (n = 34), and pathological downstaging to pTa or pT0 was required to achieve definitive CR. Pre- and post-treatment pathological staging is described in Supplementary Data 1. Patients were selected to represent all response groups. Summarized clinical, histopathological and treatment information is available in Supplementary Tables 1 and 2. Treatment response was observed for ~57% of patients (n = 172/300, NAC: ~63%, first-line: ~55%). Complete downstaging (pT0N0) following NAC treatment was observed in 52% of cases. NAC treated cases represent a very homogeneous cohort based on pre-therapeutic T-stage, and the administered chemotherapy regimen (Supplementary Table 2), which could explain the difference in observed treatment response compared to previous studies[35–37].

Informed written consent to take part in future research projects was obtained from all patients, and the specific project was approved by the National Committee on Health Research Ethics (#1706291). Biological specimens from TURB-T or metastatic lesions collected between 1995 and 2017 were provided by the Departments of Urology, Aarhus University Hospital and Departments of Pathology at Aarhus University Hospital, Randers Hospital, Aalborg University Hospital, Viborg/Holstebro Hospital, Sønderborg Hospital, Vejle Hospital, and Esbjerg Hospital. Study data were collected and managed using REDCap electronic data capture tools hosted at Aarhus University[38].

**Tissue microarrays**. Tissue microarrays (TMA) from three MIBC cohorts were included in the study, representing 184 patients in total. The TMAs consisted of up to six 1.0 mm core biopsies per patient. All core biopsies were taken from the most representative tumor area, selected by a trained pathologist.

**Nucleotide extraction procedures**. Tumor tissue was snap frozen in liquid nitrogen and stored at −80 °C or was formalin fixed and paraffin embedded (FFPE). Hematoxylin and eosin stained overview sections (top and bottom) were evaluated for the presence of carcinoma cells. DNA was extracted using the Puregene DNA purification kit (Gentra Systems) or using the QIAamp DNA FFPE tissue kit (Qiagen). DNA was extracted from peripheral blood leukocytes from all patients using the QIAsymphony DSP DNA midi kit (QIAGEN, cat#937255). RNA was quantified using an Infinite 200 PRO NanoQuant spectrophotometer (Tecan). RNA integrity was assessed using a 2100 Bioanalyzer (Agilent Technologies).

**Whole exome sequencing**. Libraries of tumor and matching germline DNA were prepared using 100–500 ng DNA and processed using the pipeline available from Roche Nimblegen (see Supplementary Data 3 for details). DNA concentration was calculated using a Qubit 3.0 fluorometer (ThermoFisher). WES library construction was made using the KAPA Hyper Library Kit (Kapa Biosystems, Roche). The libraries were sequenced (Paired end 2 × 75 bp or 2 × 150) using the Illumina NextSeq 500 platform. Due to limited DNA yield from metastatic samples, WES for these was performed using 50 ng DNA and the Twist Enzymatic Fragmentation Library prep and Human Core Exome Capture kit (Twist Bioscience, PN 100803). Raw sequencing data was initially processed using bcl2fastq2 and Trim Galore!. FastQ files were processed according to the GATK Best Practices: Alignment using bwa-mem, marking of duplicate reads using Picard, base recalibration using

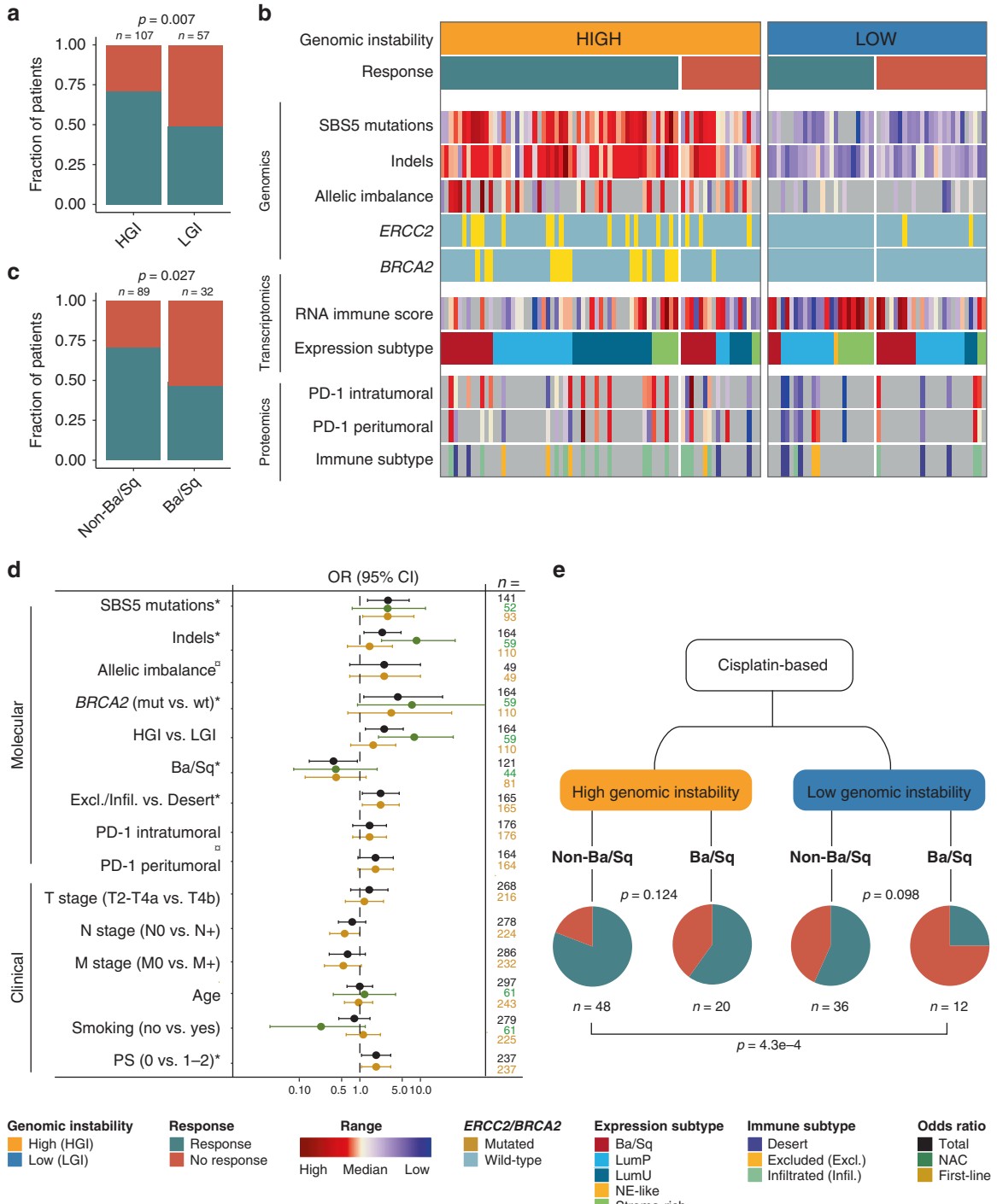

**Fig. 7 Integrative analysis. a** Chemotherapy response stratified by patient groups of high genomic instability (HGI) or low genomic instability (LGI).
**b** Integration of genomic, transcriptomic and proteomic data for all patients with genomic data available and a gene expression subtype assigned ($n = 121$),
stratified by high and low genomic instability. **c** Chemotherapy response for patients with Ba/Sq gene expression subtype and patients with non-Ba/Sq
gene expression subtype (i.e., LumP, LumU, NE-like or Stroma-rich). **d** Overview of odds ratios (OR) calculated for molecular and clinical features for all
patients (black), NAC (green) and first-line treated patients (yellow). Continuous variables were dichotomized based on the median and high vs. low is
presented. Dots indicate odds ratios and horizontal lines show 95% confidence intervals (CI). Open lines indicate that the full range of the 95% CI is not
shown. Asterisks denote p-values below 0.05. Currency signs denote p-values below 0.05 in a logistic regression model using continuous data. **e** Likelihood
of cisplatin-based chemotherapy response stratified by genomic instability and gene expression subtypes. P-values were calculated using a chi-square test.
Source data are provided as a Source data file.

GATK, quality metrics were assessed using Picard. Mutations were identified using MuTect2 with default parameters except the threshold for maximum alternate alleles in the germline was raised. A custom filter selecting variants only vastly more present in the tumor and in regions with low noise, was subsequently applied[39]. Furthermore, variants identified by MuTect2 that did not pass the built-in filters were reintroduced if they were identified with high confidence using VarScan[40] (pileups generated using samtools[41]). All somatic alterations were annotated using SnpEff[21] and hg19 build. We used the four predicted impact categories defined by SnpEff to filter alterations with high impact (frameshift variant, start codon lost or stop codon gained or lost, etc) or moderate impact

(missense variation, inframe insertion or deletion, etc) from low impact (mainly synonymous variant) and modifier impact (intergenic or intron variant, etc.).

**Neoantigen prediction.** Polysolver[42] was used to predict germline alleles of class I HLA-A, B, and C genes from the WES of the germline samples. The MuPeXI webserver[43] was then used to extract 8 to 11 length peptides around missense mutations, indels and frameshift mutations from the somatic VCF files and all mutant peptides with a binding prediction to MHC below 2% (weak binders) from NetMHCpan[44] were retained as neoantigens.

**Copy number alterations.** Custom Illumina SNP arrays (~760k positions) were produced for 49 patients (tumor + germline DNA) in order to assess copy number alterations. Genotyping, logR Ratio (LRR) and B-allele-fraction (BAF) were corrected and normalized using the Genotyping module from GenomeStudio 2.0 (Illumina) and all positions with cluster separation >0.75 were exported (594k SNPs) for further analysis. The R package ASCAT[45] was used for segmentation of the genome but various tumor DNA purities together with high heterogeneity made it difficult to obtain reliable somatic copy number estimates. Therefore, only the raw-segmented BAF data was used to define genomic regions with allelic imbalance. A sample-specific threshold was defined corresponding to a third of the max BAF segmented value (removing outliers). Regions with strong imbalance reflect genomic regions with either a loss of heterozygosity due to a loss allele (copy number loss) or due to an amplification of only one allele (imbalanced copy number gain) in most of the tumor cells.

**Mutational signature analysis and microsatellite instability.** SNVs and their trinucleotide context were subjected to mutational signature analysis using R packages SomaticSignatures[46] and MutationalPatterns[47]. Only samples with more than 100 SNVs were included to ensure robustness of the signature decomposition (162/165 samples). Trinucleotide patterns for COSMIC signatures (v3) were obtained and used for analysis of the contribution of the four previously defined BC associated signatures[48]. Given the cohort size of previous studies of mutational signatures and the consistency for mutational signatures identified in BC, we employed this approach instead of performing de novo deconvolution. To ensure the resulting contribution of mutational signatures was representative of the observed mutational spectrum, the resulting trinucleotide mutational profile for every sample was compared to the original profile and only samples with a cosine similarity above 0.9 were considered (142/162 samples). Mutational signature-based clusters were defined based on the dominant signature—SBS1, SBS5 or the APOBEC signatures (SBS2 and 13). For volcano plots, only genes mutated in more than 5% of the relevant cohort were considered. Only mutations with high or moderate impact (based on SnpEff annotation) on the resulting protein were included in this analysis. To account for varying mutation burdens across samples and genes, a permutation test[20] was applied to assess significance. Initially, p-values (p_obs) were calculated for every gene using a Wilcoxon rank-sum test. A matrix consisting of samples as columns and genes as rows and the appropriate mutation status inserted, was generated and permuted using the curve_ball function presented by Strona et al.[49] P-values (p_perm) were then calculated using a Wilcoxon rank-sum test for every gene in every permuted matrix and the final p-value was calculated as the fraction of p_perm smaller than p_obs. TCGA data were processed similarly for analyses related to mutational signatures. MSI status was inferred using MSIsensor[50].

**Damaging DNA damage response mutations.** Missense mutations in genes related to DDR were analyzed for potentially damaging effects using Polyphen2[18] and MutationAssessor[19]. Mutations identified as possibly damaging/probably damaging or medium/high in PolyPhen2 and MutationAssessor, respectively, were considered damaging. All loss of function mutations were considered damaging.

**Comparison of primary tumors and metastatic lesions.** For each patient, we called mutations using MuTect2 (default parameters) by comparing all available samples (primary tumor and metastatic lesions) to the germline. However, a mutation might be called in e.g. the primary tumor and not quite reach the threshold of being called in a metastatic lesion. We, therefore, combined all called mutations for each patient and assessed the presence of said mutations in all relevant bam files using bam-readcount (only reads and bases with high quality were considered). Based on the minimum observed VAF for a mutation across all relevant samples, we calculated the minimum required read depth to observe it and kept only mutations with sufficient read depth in all relevant samples.

**RNA sequencing.** RNA-seq was performed using the QuantSeq kit FWD HT kit (Lexogen) using 500 ng input RNA. The method generates 3′ count data from polyadenylated RNA and only one fragment is assessed per transcript ensuring highly accurate gene expression values. The process was carried out according to the manufacturer for RNA procured from frozen and FFPE material. The QuantSeq libraries were sequenced (Single end 1 × 75 bp) using the Illumina NextSeq 500 platform.Salmon[51] was used to quantify the amount of each transcript using annotation from GRCh38 for cDNA and non-coding RNA (ncRNA). The R-

packages, tximport[52] and edgeR[53], were used to respectively summarize the expression at gene-level and normalize the data.

Samples were classified according to the six consensus classes of MIBC using the R-based consensus MIBC classification tool ($n = 121$)[15]. All samples were used for the expression subtyping but only the fresh frozen samples ($n = 96$) were used for further analysis.

**Analysis of regulon activity.** We investigated transcriptional regulatory networks consisting of transcription factors and associated induced/repressed target genes using R package RTN[54]. Our analysis was confined to 23 transcription factors previously associated with BC (Supplementary Table 4)[5,55]. In brief, a normalized gene expression matrix was used to infer associations between transcription factors and regulated target genes. Unstable associations were removed by bootstrap analysis. A gene can be associated with more than one transcription factor at this point and the weakest interaction was, therefore, removed. Finally, the activity of all regulons was assessed using a two-tailed gene set enrichment analysis. We furthermore performed cell type enrichment analysis using xCell with the same normalized gene expression matrix as above as input[56].

**Immune cell deconvolution.** The presence of immune cells was evaluated based on the expression of predefined gene lists for every immune cell type of interest[57] (for CD4 cells[58]). A score for every cell type was calculated as the mean expression for all genes associated with the cell type in question. An overall immune score was defined as the sum of all immune cell type scores.

**DNA methylation.** DNA methylation analysis was performed using DNA from 72 patients. We used 500 ng genomic DNA for bisulfite conversion followed by whole-genome amplification prior to hybridization to EPIC BeadChip (Illumina, San Diego, CA) overnight as described by the manufacturer and then scanned with the Illumina iSCAN system. Data were imported and normalized using the ChAMP R package[59]. Methylation microarrays data (450k) was retrieved from the TCGA project for normal bladder tissue and from[60] for leukocytes. We identified 5000 tumor-associated hyper- and 5000 hypomethylated CpG sites[5]. For the tumor-associated hypermethylated sites, we selected 5000 CpG autosomal sites unmethylated in normal bladder tissue and in leukocytes, present on the EPIC platform and methylated in at least four patients. Unsupervised consensus clustering[61] was used with each set and for both cases, three major clusters were found.

All probes annotated as TSS1500, TSS200 or 5′UTR were summarized to define the promoter methylation value and all probes annotated as Body were summarized to define the gene body methylation value for all genes.

**Immunofluorescence, immunohistochemistry and imaging.** Two TMA sections per patient (3 μm) were analyzed using multiplex immunofluorescence to detect the expression of markers in Panel 1 (CD3, CD8, FOXP3) and Panel 2 (CD20, CD68, CD163, HLA-ABC). Primary antibodies are listed in Supplementary Data 2. Staining was carried out on the Discovery ULTRA Staining instrument (Ventana Medical Systems) using Ventana Medical Systems reagents except as noted. TMA sections were deparaffinized using EZ Prep solution (cat#950-102) for 16 min at 72 °C, followed by heat-induced epitope retrieval for antigen unmasking with CC1 solution (cat#950-124) at high temperature (e.g., 95–100 °C) for 64 min. Endogenous peroxidase activity was blocked using DISC inhibitor reagent (cat#760-4840). For fluorescent detection, the first primary antibody was incubated followed by detection using a goat anti-mouse (GaM-HRP, 12 min; OmniMap anti-Ms HRP (RUO), DISCOVERY; cat#760-4310) or goat anti-rabbit (GaR-HRP; 16 min; OmniMap anti-Rs HRP (RUO), DISCOVERY; cat#760-4311) secondary antibody conjugated to horseradish peroxidase (HRP). After two rinses with reaction buffer (cat#950-300), the appropriate tyramide conjugated fluorophore (Ty-flour, listed in Supplementary Data 2) was added for 4 min with the application of 0.01% $H_2O_2$ (DISCOVERY reagent; cat#760-244). For the following 8 min the Ty-flour reacts with the HRP in the primary antibody/secondary antibody complex, which leads to oxidation and subsequent covalent binding of Ty-fluor to tyrosine residues surrounding the antigens. Finally, we performed a heat-mediated stripping of the antibodies (100 °C for 20 min in CC2 buffer, cat#950-223). The above processing cycle was repeated sequentially two (Panel 1) or three (Panel 2) more times using a different antibody and fluorophore. VECTAshield anti-fade mounting medium with DAPI (cat#H-1200) was added as a nuclear counterstain. Following fluorophore imaging using the Hamamatsu NanoZoomer s60 scanner (Meyers Instruments), immunostaining on the same section was performed to study spatial organization. Intratumoral regions were identified using the antibody PAN-cytokeratin (Clone A1/A3, 1:100; 16 min; Dako Agilent; cat#GA005361-2). Additionally, three TMA sections per patient were obtained and immunohistochemistry was performed with antibodies directed against PD-1(Clone NAT105; ready-to-use; 32 min; cat#760-4895), PD-L1(Clone NAT105; ready-to-use, 60 min; cat#790-4905) and PAN-cytokeratin (Clone A1/A3, 1:100; 16 min; Dako Agilent; cat#GA005361-2). For bright-field detection, slides were developed using the Ventanas Detection Kits: ultraView Universal 3,3′-Diaminobenzidin (cat#760-500) according to the manufacturer's instructions. Slides were then counterstained with hematoxylin II (cat#790-2208) for 8 min, followed by Bluing reagent (cat#760-

2037) for 4 min. Bright-field imaging was performed in the Hamamatsu Nano-zoomer 2.0 HT (Meyers Instruments).

**Automated quantification of scanned images**. Automated quantification of selected markers was carried out using the Visiopharm software (Visiopharm A/S, Hørsholm, Denmark). For each tissue core, the fluorescence image was aligned to its corresponding chromogenic (cytokeratin) stained image using the Visiopharm Tissue Align module (Supplementary Fig. 7a). Image analysis protocol packages (APPs) were developed in order to automatically 1) Define intratumoral and peritumoral regions of interest (Supplementary Fig. 7b), 2) Calculate fractions of immune cell subsets based on co-localization of selected markers (Supplementary Fig. 7c), 3) Calculate H-score [$1 \times$ (% cells low intensity) + $2 \times$ (% cells moderate intensity) + $3 \times$ (% cells high intensity)] for MHC-expression on carcinoma cells. 4) Calculate fractions of PD-1/PD-L1 positive cells. For panel 2 and PD-1 staining, the intratumoral and peritumoral regions of interest were manually defined by visual inspections. For each selected marker, a visually defined threshold was set to differentiate between positive and unspecific staining. The following scoring algorithms were applied to calculate cell fractions (here shown for CTLs (CD8 + CD3 + FOXP3-) as an example):

Intratumoral CTLs = CTLs counts intratumoral/total cell count intratumoral
Peritumoral CTLs = CTLs counts peritumoral/total cell count peritumoral.

**Immune subtypes and PD-1/PD-L1 axis**. Immune Scores (IS) for each region of interest were calculated based on the sum of immune cell fractions (T-Helper, CTLs, Tregs, B-cells, M1 and M2 macrophages) present in the specific region. The IS were then split at the median to form high and low groups. We classified our three immune subtypes as follows: (1) Desert: Intratumoral $IS_{low}$ + Peritumoral $IS_{low}$, (2) Excluded: Intratumoral $IS_{low}$ + Peritumoral $IS_{high}$, (3) Infiltrated: Intratumoral $IS_{high}$ + Peritumoral $IS_{high}$ or Intratumoral $IS_{high}$ + Peritumoral $IS_{low}$. For evaluation of the combined PD-1/PD-L1 axis (Fig. 6h), samples were dichotomized as positive and negative by the median.

**Statistics and reproducibility**. All multi-omics measurements were performed once for each distinct sample. Assessment of statistical significance included Fisher's exact test or chi-square test for categorical variables and Wilcoxon rank-sum test for continuous variables. Significance levels were adjusted for multiple testing using the Bonferroni method for Figs. 4d, h, 5c, 6I, j. Cumulative survival analysis was performed using the Kaplan–Meier method, and the log-rank test was used to compare the curves (R packages survminer and survival). Overall survival (OS) was defined as time from MIBC diagnosis to death or end of follow-up. Eight patients received both NAC and first-line treatment for metastatic disease, in these cases, the first-line response evaluation was used for statistical assessment of treatment response. Genomic instability groups were defined by the number of SBS5 mutations, indels and allelic imbalance and *BRCA2* mutation status. If patients had numbers above the median for SBS5 mutations, indels or allelic imbalance or a *BRCA2* mutation, they were assigned to the group of high genomic instability. The protein immune score (IS), defined as the sum of immune cell fractions, was dichotomized based on the median.

**Reporting summary**. Further information on research design is available in the Nature Research Reporting Summary linked to this article.

## Data availability

Raw sequencing, SNP microarray and methylation data are deposited and available under controlled access at the European Genome-phenome Archive (EGA), which is hosted by the European Bioinformatics Institute (EBI) and the Centre for Genomic Regulation (CRG). Study accession numbers are: EGAS00001004507 (WES), EGAS00001004519 (SNP data), EGAS00001004505 (RNA-Seq), and EGAS00001004515 (EPIC BeadChip methylation data). Normalized mRNA read counts are available in Supplementary Data 4. Source data are provided with this paper. TCGA WES, methylation and clinical data was accessed at [https://portal.gdc.cancer.gov/projects/TCGA-BLCA]. 450k methylation data for leukocytes were retrieved from the Gene Expression Omnibus (Series GSE32148). Only samples annotated as normal peripheral blood were used. The remaining data are available within the Article file, Supplementary Information or available from the authors upon request. Source data are provided with this paper.

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

## Acknowledgements

We thank Nicolai Birkbak for constructive feedback on the manuscript, Lasse Maretty Sørensen for statistical assistance, and Jeanette Bæhr Georgsen for technical assistance regarding the proteomics analysis. The results presented here are in part based upon data generated by the TCGA Research Network: http://cancergenome.nih.gov/. This work was funded by the Novo Nordisk Foundation (NNF17OC0024464), Independent Research Fund Denmark (7016-00369b), Aarhus University, The Danish Cancer Biobank, KV Fonden, Toyota-Fonden (Denmark) and Beckett-Fonden.

## Author contributions

Conceptualization, A.T. and L.D.; Methodology, A.T., E.C., P.L., I.N., F.F.P., T.L.H.O., M.K., J.S.P., and L.D.; Formal analysis, A.T., E.C., and P.L.; Investigation, A.T., E.C., P.L., S.V.L., and L.D.; Resources, A.T., K.B., M.A., and J.B.J.; Data curation, P.L. and M.K.; Writing—original draft, A.T., E.C., P.L., and L.D.; Writing—review and editing, A.T., E.C., P.L., I.N., F.F.P., S.V.L., K.B., T.L.H.O., M.K., J.S.P., T.S., M.A., J.B.J., and L.D.; Supervision, L.D.; Project administration, L.D.; Funding acquisition, A.T. and L.D.

## Competing interests

L.D. has sponsored research agreements with C2i genomics, AstraZeneca, Natera and Ferring, and has an advisory/consulting role at Ferring. J.B.J. is proctor for Intuitive Surgery, member of advisory board for Olympus Europe, Cephaid and Ferring, and has sponsored research agreements with Medac, Photocure ASA, Cephaid and Ferring. The following authors declare no competing interests: A.T., E.C., P.L., I.N., F.F.P., S.V.L., K.B., T.L.H.O., M.K., J.S.P., M.A., and T.S.

## Additional information

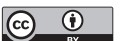

