## [Peer Review File · Nature Communications]

REVIEWER COMMENTS

Reviewer #1 (Remarks to the Author): Expert in bladder cancer

Taber et al report genomic and molecular characterization of 300 bladder cancer samples and correlate findings with response to chemotherapy. The study provides additional data to address the important goal of identifying determinants of response to chemotherapy. Some of the conclusions are add odds with published work as discussed by the authors. While unfortunately these conflicting results raise new questions, they are important to share with the research community.

The primary value of the publication is as a data resource. As such it is critical the investigators provide a well annotated spreadsheet along with the publication such that others can compare results to other studies.

For each sample, it should be clear:

1. Stage of tumor at diagnosis
2. Method for assessing response/no response – RECIST vs pathologic stage. If pathologic stage, stage post-therapy should also be provided.
3. Specifics about chemotherapy – number of cycles successfully completed and specific agents
4. All info in Figure 1A should be provide in table format. This data should be extended to include all genes found to be significantly mutated or to have significant CNV in bladder cancer by the TCGA.

Similarly, a major difference with this study and previous work is the high proportion of patients being treated with metastatic disease. It would be of interest to report how the -omics variables analyzed predict response only in the NAC group since previous work has been primarily in the NAC setting.

Minor comments:

1. The authors suggest they are measuring chemotherapy response. When using pathologic staging from a TURBT specimen, it is important to make clear that down-staging may not be due to chemotherapy response, but simply may reflect down-staging from a previous TURBT. Factors that drive a slow natural history of disease could score as sensitizers for chemotherapy response, when in fact they simply predict low likelihood of recurrence post TURBT. Based on a controlled neo-adjuvant study in which a 15% pT0 rate was found even in the arm that did not receive NAC (N Engl J Med 2003; 349:859-866), it is clear that down-staging is not only the consequence of chemotherapy response.

2. The response rate, particularly in the NAC setting is surprisingly high at 66%. Do the authors have any thoughts about why the response rate is higher than expected compared to older studies (e.g. Cancer. 2008 Nov 1;113(9):2471-7)

Reviewer #2 (Remarks to the Author): Expert in bladder cancer

Besides cystectomy, the current mangement of MIBC includes the neoadjuvant or adjuvant platin–

based chemotherapy. This leads in many cases to patient overtreatment without improvement of clinical response. Accordingly, a possible patient stratification and the suitable identification of biomarkers of response or resistance are a current need in the management of bladder cancer patients.

Here Taber et al., describes a multiomic characterization of an ample series of neo- and adjuvant cisplatin-treated patients. They characterize that genomic instability and specific mutational signature, along with Immune cell infiltration and high PD-1 protein expression were associated with response. On the other hand expression data identified the basal/squamous subtype to be associated with poor response. The integration of genomic and transcriptomic data allow the patient stratification into low and high likelihood of response.

The work is adequately performed, manuscript is clearly written and the conclusions are supported by the data and their analyses. A better clarification regarding neoadjuvant or adjuvant treatment in the patient series and the identification of potential differences between these two groups is strongly recommended. A potential problem is the use of the selected biomarkers in a prospective validation dataset. Undoubtedly this will be a possible future work of this and other group working on this and related aspects of BC management.

Reviewer #4 (Remarks to the Author): Expert in computational biology

In the current manuscript, Taber et al. presented a comprehensive analysis of a large cohort of muscle-invasive bladder cancers treated with cisplatin, to develop a novel patient stratification strategy that is predictive for chemotherapy response. Potential molecular markers were systematically screened on multiple levels, including somatic mutations and LOH, gene expression patterns, methylation features, and intra-/peri-tumoral immune profiles. The somatic alterations were summarized to extract overall sample-level patterns, such as trinucleotide mutational signature, mutational burden and genomic instability. The mutational signature was also used as one of the major classifiers of tumor subgroups. The genomic analysis was further extended to a subset of primary/metastatic pairs to characterize intertumoral heterogeneity. Through these analyses, they identified several molecular markers associated with clinical response, including high genomic instability, mutational signature #5, presence of BRCA2 mutations, Ba/Sq expression type, immune activity, including immune infiltration and PD-1 expression. By integrating these markers, they were able to identify the subgroups of patients with very high (80%) and low (25%) response rates. Overall, the current work represents a comprehensive effort for identifying novel molecular biomarkers for improving patient stratification strategy in MIBC. Although the study is well-designed, the results look solid, and the paper appears to be well-written, several questions could be better addressed to improve the current draft:

Major:

1. One important piece of analysis that might be added to the current analysis: how was the neoantigen burden, as introduced by SNVs and Indels, correlated with treatment response under each circumstance? Although the neoantigen burden is expected to be correlated with mutation load in general, it may provide better predictive value. The authors found that “immune infiltration per se was not associated with response to chemotherapy”. Would the combination of neoantigen

burden and immune infiltration provide a better association with the treatment response?

Minor:

1. The Indel burden was found to be associated with response. For the patients with high Indels, how many of them are caused by MSI? Could MSI be a better predictor?
2. For the description of sequencing coverage depth, more details would be helpful. For example, what percent of coding regions are covered by at least the desired coverage, e.x. 20X, in each sample?
3. In the comparison of primary vs. met in figure 3c, much fewer primary-specific mutations were identified, compared with metastatic-specific mutations. Was there any systematic difference between the primary and met tumors in other factors, specifically, tumor purity, and sequencing statistics, especially since they were sequenced in different batches? What was the minimum threshold for defining a mutation being absent in a particular sample?
4. An important finding is the presence of somatic BRCA2 was associated with treatment response. For the remaining patients without somatic BRCA2, could any of them harbor germline BRCA2 deleterious variants? If so, it may further improve the association.
5. In the pairwise comparison, such as Figure 4d, were the p values adjusted for multiple testing correction?
6. In the methods, it was indicated that WES was analyzed in hg19, but RNASeq in GRCh38. It may not be critical but was there any particular reason for using two different reference assemblies?

Response to reviewers

Reviewer #1 (Remarks to the Author): Expert in bladder cancer

Taber et al report genomic and molecular characterization of 300 bladder cancer samples and correlate findings with response to chemotherapy. The study provides additional data to address the important goal of identifying determinants of response to chemotherapy. Some of the conclusions are add odds with published work as discussed by the authors. While unfortunately these conflicting results raise new questions, they are important to share with the research community.

The primary value of the publication is as a data resource. As such it is critical the investigators provide a well annotated spreadsheet along with the publication such that others can compare results to other studies.

For each sample, it should be clear:

1. Stage of tumor at diagnosis
2. Method for assessing response/no response – RECIST vs pathologic stage. If pathologic stage, stage post-therapy should also be provided.
3. Specifics about chemotherapy – number of cycles successfully completed and specific agents

Author response: We thank the reviewer for the excellent suggestions. Below follows answers to the points raised:

1. T-stage at diagnosis has now been added to **Fig. 1A**, and **Supplementary Table 1** includes diagnostic TNM stage in a table format for all patients. Furthermore, we have added diagnostic TNM-stage information to **Supplementary table 2** and **3** summarizing clinical characteristics and multi-omics platforms and treatment regimes (NAC and First-line), respectively.
2. Treatment response evaluation is critical to our manuscript and we appreciate the opportunity to make this more clear. The following description have been included in the methods section regarding patients details:

*“ Pretreatment staging was based on cross-sectional imaging (baseline) and pathological assessment of TURB-T (transurethral resection of bladder tumor) specimen. NAC treatment response was defined as pathological noninvasive downstaging ($\leq pTa, CIS, N0$) based on examination of the CX specimen. First-line treatment response was defined as complete (CR) or partial response (PR) based on post-treatment cross-sectional imaging according to the RECIST 1.1 guidelines (Response Evaluation Criteria in Solid Tumors)³³. 55.5% (136/245 patients) had an intact bladder at the time of first-line treatment. Posttreatment pathological staging of the residual tumor was evaluated in patients with radiologic CR (N=34), and pathological downstaging to pTa or pT0 was required to achieve definitive CR. Pre- and posttreatment pathological staging is described in **Supplementary Table 1**. ”*

Furthermore, **Supplementary Table 1** has been expanded, and now gives a full description of pre- and post pathological staging.

3. In **Supplementary Table 1** we have provided specific details for all patients regarding treatment regimes, agents, number of successfully completed cycles and treatment response data.

Reviewer 1:

All info in Figure 1A should be provide in table format. This data should be extended to include all genes found to be significantly mutated or to have significant CNV in bladder cancer by the TCGA.

Author response:

In order to maximize the availability of our research data as a data source for other studies, we now include a **Source Data** file as recommended by Nature Communications. The **Source Data** file includes all data for each figure provided in a separate sheet, including Figure 1A in the sheet named "Figure 1". Due to different data formats and file sizes, source data for the following figures is available as separate files: Figure 2c+d, Figure 3, Figure 5, Figure 7d, Supplementary Figure 4b.

We are committed to making the data easily available and have therefore, in addition to the source data (Data File 1), included a table with normalized expression data (Data File 2).

We have furthermore uploaded whole exome sequencing, RNA sequencing, methylation, and SNP data to the European Genome-Phenome Archive, as described in the data availability section.

Reviewer 1: Similarly, a major difference with this study and previous work is the high proportion of patients being treated with metastatic disease. It would be of interest to report how the -omics variables analyzed predict response only in the NAC group since previous work has been primarily in the NAC setting.

Author response:

We agree with the reviewer this is important to highlight. We show odds-ratios for omics variables for the NAC and first line cohorts separately in Figure 7d. However, we acknowledge this was not sufficiently clear. We have therefore added the following sentence to the legend of Figure 7d:

"d. Overview of odds ratios (OR) calculated for molecular and clinical features for all patients (black), NAC (green) and first-line treated patients (yellow)."

Furthermore, we have now grouped patients based on genomic instability and gene expression subtype, similarly to Figure 7e, but for NAC treated and first-line treated patients separately. This is now presented in **Supplementary Fig. 8**. Importantly, we see the same pattern as for the combined cohort. We have made the following changes to the results section to describe this:

"Importantly, this combined analysis identifies a group of patients with a very high response rate (80%, NAC: 90%, First-line: 71%) characterized by high genomic instability and non-Ba/Sq gene expression subtype and a group of patients with a very low response rate (25%, NAC: 20%, First-line: 29%) characterized by low genomic instability and Ba/Sq gene expression subtype ($p < 0.001$, NAC: $p = 0.006$, First-line: $p = 0.077$) (Supplementary Fig. 8)."

Reviewer 1:**Minor comments:**

1. The authors suggest they are measuring chemotherapy response. When using pathologic staging from a TRUBT specimen, it is important to make clear that down-staging may not be due to chemotherapy response, but simply may reflect down-staging from a previous TURBT. Factors that drive a slow natural history of disease could score as sensitizers for chemotherapy response, when in fact they simply predict low likelihood of recurrence post TURBT. Based on a controlled neo-adjuvant study in which a 15% pT0 rate was found even in the arm that did not receive NAC (N Engl J Med 2003; 349:859-866), it is clear that down-staging is not only the consequence of chemotherapy response.

Author response:

We agree with the reviewer that it is important to acknowledge TURB-T prior to NAC may influence the downstaging rate. We have therefore now included the following statement in the discussion section:

“Although pathological downstaging has been used for evaluating NAC response in several studies 2,3, it is not without limitation, including the potential impact of previous TURB-T on the rate of downstaging”

Reviewer 1:

The response rate, particularly in the NAC setting is surprisingly high at 66%. Do the authors have any thoughts about why the response rate is higher than expected compared to older studies (e.g. Cancer. 2008 Nov 1;113(9):2471-7).

Author response: We thank the reviewer for the reflections and we agree that the response rate in the NAC cases is high (N=39/62, 63%) compared to previous findings, where the extent of pT0 following NAC varies between 14% and 38% ⁴⁻⁶. In our study, NAC response was defined as pathological **noninvasive** downstaging (\leq pTa,CIS, N0). Complete downstaging (pT0N0) was observed in 52% of cases. As we are interested in identifying biological correlates to pathological downstaging and response, we used the less conservative definition of downstaging to noninvasive disease. Factors that may influence the rate of downstaging in other reported studies include diagnostic T-stage, use of different treatment regimes, no. of completed series and use of local radiation. In our study 56 of 62 cases had pre-therapeutic T2 stage (55% had pre-therapeutic T3-4 stage in Cancer. 2008 Nov 1;113(9):2471-7), 61 of 62 cases received cisplatin and gemcitabine, 53 of 62 cases completed ≥ 3 series, and none had local radiation. Our NAC cases represent a very homogeneous cohort, which could explain the difference in observed treatment response compared to previous studies. We have added the following sentence to the method section *Patient Details:*

*“NAC treated cases represent a very homogeneous cohort based on pre-therapeutic T-stage, and the administered chemotherapy regimen (**Supplementary Table 3**), which could explain the difference in observed treatment response compared to previous studies”*

Reviewer #2 (Remarks to the Author): Expert in bladder cancer

Besides cystectomy, the current management of MIBC includes the neoadjuvant or adjuvant platinum-based chemotherapy. This leads in many cases to patient overtreatment without improvement of

clinical response. Accordingly, a possible patient stratification and the suitable identification of biomarkers of response or resistance are a current need in the management of bladder cancer patients.

Here Taber et al., describes a multiomic characterization of an ample series of neo- and adjuvant cisplatin-treated patients. They characterize that genomic instability and specific mutational signature, along with Immune cell infiltration and high PD-1 protein expression were associated with response. On the other hand expression data identified the basal/squamous subtype to be associated with poor response. The integration of genomic and transcriptomic data allow the patient stratification into low and high likelihood of response.

The work is adequately performed, manuscript is clearly written and the conclusions are supported by the data and their analyses.

A better clarification regarding neoadjuvant or adjuvant treatment in the patient series and the identification of potential differences between these two groups is strongly recommended.

Author response: We thank the reviewer for the kind words and suggestions for improvement. We acknowledge clarification was required regarding chemotherapy regimens. We have therefore now included **Supplementary Table 1**, which outlines the pre- and post therapeutic staging, treatment regimes, agents, number of successfully completed cycles and treatment response data. We believe this will make the administered treatment regimens clearer to the reader.

We would also like to point out that patients in this study received either neoadjuvant chemotherapy (NAC) before cystectomy or first-line chemotherapy upon detection of locally-advanced (T4b) or metastatic disease. Seven patients received both NAC and first-line treatment. To make this more clear we made the following changes to the manuscript:

- In the abstract: *“Here we performed a comprehensive multi-omics analysis (genomics, transcriptomics, epigenomics and proteomics) of 300 MIBC patients treated with chemotherapy (neoadjuvant or first-line) to identify molecular changes associated with treatment response.”*
- Result section: *“To investigate molecular correlates associated with treatment response, we included 300 tumors from patients with BC receiving chemotherapy; 62 received NAC before cystectomy (CX) and 245 received first-line chemotherapy upon detection of locally-advanced (T4b) or metastatic disease (Supplementary Fig. 1a).”*

To address potential differences between the cohorts (NAC and first-line) in this study we presented odds-ratios for omics variables for the NAC and first line cohorts separately in Figure 7d. However, we acknowledge this was not sufficiently clear. We have therefore added the following sentence to the legend of Figure. 7d:

“d. Overview of odds ratios (OR) calculated for molecular and clinical features for all patients (black), NAC (green) and first-line treated patients (yellow).”

Furthermore, we have now grouped patients based on genomic instability and gene expression subtype, similarly to Figure 7e, but for NAC treated and first-line treated patients separately. This is now presented in **Supplementary Fig. 8**. Importantly, we see the same pattern as for the combined cohort. We have made the following changes to the results section to describe this:

“Importantly, this combined analysis identifies a group of patients with a very high response rate (80%, NAC: 90%, First-line: 71%) characterized by high genomic instability and non-Ba/Sq gene expression subtype and a group of patients with a very low response rate (25%, NAC: 20%, First-line: 29%) characterized by low genomic instability and Ba/Sq gene expression subtype ($p < 0.001$, NAC: $p = 0.006$, First-line: $p = 0.077$) (Supplementary Fig. 8).”

Reviewer 2:

A potential problem is the use of the selected biomarkers in a prospective validation dataset. Undoubtedly this will be a possible future work of this and other groups working on this and related aspects of BC management.

Author response: We agree that validation in future prospective studies is an important next step for our reported biomarkers. We mention this as a final remark in the manuscript:

“If successfully validated in future prospective trials, these findings could aid in selecting patients with a high probability of treatment response and potentially minimize the current overtreatment of patients. Prospective validation is currently ongoing (NCT04138628).”

As mentioned we have initiated a multicenter prospective study in 2020 (NCT04138628). We are currently recruiting patients and, hence data for validation of the present study are not yet available. To the best of our knowledge, our work is the first study correlating multi omics data to chemotherapy response in a clinically well-annotated cohort of patients with bladder cancer - and consequently, utilization of publically available data for further verification is currently not possible. However, as also stated by reviewer 1, this work will also serve as a data resource for other studies investigating chemotherapy response.

Reviewer #4 (Remarks to the Author): Expert in computational biology

In the current manuscript, Taber et al. presented a comprehensive analysis of a large cohort of muscle-invasive bladder cancers treated with cisplatin, to develop a novel patient stratification strategy that is predictive for chemotherapy response. Potential molecular markers were systematically screened on multiple levels, including somatic mutations and LOH, gene expression patterns, methylation features, and intra-/peri-tumoral immune profiles. The somatic alterations were summarized to extract overall sample-level patterns, such as trinucleotide mutational signature, mutational burden and genomic instability. The mutational signature was also used as one of the major classifiers of tumor subgroups. The genomic analysis was further extended to a subset of primary/metastatic pairs to characterize intertumoral heterogeneity. Through these analyses, they identified several molecular markers associated with clinical response, including high genomic instability, mutational signature #5, presence of BRCA2 mutations, Ba/Sq expression type, immune activity, including immune infiltration and PD-1 expression. By integrating these markers, they were able to identify the subgroups of patients with very high (80%) and low (25%) response rates. Overall, the current work represents a comprehensive effort for identifying novel molecular biomarkers for improving patient stratification strategy in MIBC. Although the study is well-designed, the results look solid, and the paper appears to be well-written, several questions could be better addressed to improve the current draft:

Major:

1. One important piece of analysis that might be added to the current analysis: how was the neoantigen burden, as introduced by SNVs and Indels, correlated with treatment response under each circumstance? Although the neoantigen burden is expected to be correlated with mutation load in general, it may provide better predictive value. The authors found that “immune infiltration per se was not associated with response to chemotherapy”. Would the combination of neoantigen burden and immune infiltration provide a better association with the treatment response?

Author response:

We thank the reviewer for pointing this out. We have now assigned HLA types using Polysolver and calculated the predicted neoantigens using Mupexi (<http://www.cbs.dtu.dk/services/MuPeXI/>). As the reviewer also states, the number of neoantigens is highly correlated with the number of mutations (snvs + indels; correlation coefficient = 0.90 in our data). We have now included **Fig. 1d** which addresses the correlation with treatment response, however it was not significant. We now write the following:

“For patients responding to chemotherapy, we observed a significantly higher number of indels ($p = 0.031$; SNVs: $p = 0.38$; Neoantigens: $p = 0.17$; Fig. 1b-d) and a significantly higher proportion of the genome under allelic imbalance (SNP arrays ($N = 49$); $p = 0.024$), indicating that a more disrupted genome is more sensitive to treatment with chemotherapy (Fig. 1d).”

In addition, we now include a new **Supplementary Fig. 5b** which is based on stratification of patients based on immune infiltration and neoantigens. However, we do not observe any significant association with chemotherapy response. We have added the following sentence to the manuscript:

“Grouping patients based on immune infiltration and neoantigen load neither displayed an association with response to chemotherapy (Supplementary Fig. 5b).”

Reviewer 4:

Minor:

1. The Indel burden was found to be associated with response. For the patients with high Indels, how many of them are caused by MSI? Could MSI be a better predictor?

Author response: We agree with the reviewer that this is interesting to investigate. We have now applied the tool MSIsensor to derive the extent of MSI for all patients. We obtained the scores illustrated below:

The authors of the tool use a cutoff at 3.5 for comparisons with experimental data⁷, which for our study cohort results in 1 patient with MSI out of 165 patients in total. Based on the literature, we do expect to see few bladder tumors with MSI. A recent pan cancer paper identified 11/355 bladder tumors with MSI using MSIsensor and a cutoff at 10⁸. We therefore consider the MSI score distribution observed for our cohort to reflect an expectable result. We have added the following sentence to the results section of the manuscript:

*“To further address genome disruption, we computed the microsatellite instability (MSI) status for all patients, however MSI status was not associated with genome disruption or chemotherapy response in this study (**Supplementary Fig. 2b**).”*

And the following section to the methods section:

“MSI status was inferred using MSIsensor.”

Reviewer 4:

For the description of sequencing coverage depth, more details would be helpful. For example, what percent of coding regions are covered by at least the desired coverage, e.x. 20X, in each sample?

Author response:

The fraction of target regions covered by at least 10X, 20X, 30X, 40X, 50X and 100X is now presented for each sample in **Supplementary Table 7**. The median fraction of target bases at 20X is now written in the manuscript:

“WES was performed using DNA from 165 tumors (76x median coverage, median of 92% bases at 20X) and associated leukocyte germline DNA (46x median coverage, median of 85% bases at 20X).”

Reviewer 4:

In the comparison of primary vs. met in figure 3c, much fewer primary-specific mutations were identified, compared with metastatic-specific mutations. Was there any (FF vs FFPE, twist vs medexome) systematic difference between the primary and met tumors in other factors, specifically, tumor purity, and sequencing statistics, especially since they were sequenced in different batches? What was the minimum threshold for defining a mutation being absent in a particular sample?

Author response:

We prioritized FF tissue over FFPE for generating WES data. For all patients with associated metastatic lesions, DNA from primary tumors was extracted from FF tissue. However, for all metastatic lesions only FFPE tissue was available.

Furthermore, due to limited DNA yield from DNA extraction from metastatic lesions, we generated libraries and performed exome capture using the pipeline available from Twist BioScience instead of generating libraries using KAPA Hyper and performing exome capture using the MedExomePlus panel. We now more clearly state in the methods section why Twist instead of MedExomePlus was prioritized for metastatic samples:

“Due to limited DNA yield from metastatic samples, WES for these was performed using 50 ng DNA and the Twist Enzymatic Fragmentation Library prep and Human Core Exome Capture kit”.

The target intervals of these capture panels do, however, differ slightly. The methodology for comparing the WES data files and addressing potential systematic differences, was previously described in the legend associated with Figure 3, however to make it more clear to the reader we have now created a section describing it in the supplementary methods:

“For each patient, we called mutations using MuTect2 (default parameters) by comparing all available samples (primary tumor and metastatic lesions) to the germline. However, a mutation might be called in e.g. the primary tumor and not quite reach the threshold of being called in a metastatic lesion. We therefore combined all called mutations for each patient and assessed the presence of said mutations in all relevant bam files using bam-read count (only reads and bases with high quality were considered). Based on the minimum observed VAF for a mutation across all relevant samples, we calculated the minimum required read depth to observe it and kept only mutations with sufficient read depth in all relevant samples.”

Collectively, this ensures only statistically robust genomic positions are considered and that they are thoroughly examined for the presence of mutations. It also ensures comparability between samples is prioritized in order to limit the impact of systematic differences. We do, however, acknowledge e.g. error rates of the applied polymerases might differ between WES pipelines. To make it more clear to the reader that minor systematic differences are at play, we have added the following to the results section:

“However, this observation might be impacted by DNA from primary tumors being extracted from FF tissue and DNA from metastatic lesions being extracted from FFPE.”

And the following to the discussion:

“The usage of different WES library pipelines might have impacted this observation, however our analyses only considers genomic positions with sufficient data across all samples in order to minimize the impact of potential systematic differences.”

Reviewer 4:

An important finding is the presence of somatic BRCA2 was associated with treatment response. For the remaining patients without somatic BRCA2, could any of them harbor germline BRCA2 deleterious variants? If so, it may further improve the association.

Author response:

This is a very good point, however, in our ethics approvals for the project, which is based on bio-banked materials for future research purposes, we are not allowed to directly investigate germ line disease causing variants (like e.g. mutations in BRCA2). It is a recommendation from our national scientific ethical committee that we should avoid investigating these disease causing genes (they refer to this list: <https://www.ncbi.nlm.nih.gov/clinvar/docs/acmg/> regarding genes to avoid investigating), as identification of mutations will require genetic counseling of the affected patients and family members. Consequently, we cannot include this analysis, although it would be highly relevant and something that should be performed in prospective studies with specific informed consent to do this.

Reviewer 4:

In the pairwise comparison, such as Figure 4d, were the p values adjusted for multiple testing correction?

Author response:

We have now adjusted the displayed p-values for the following figures for multiple testing: Fig. 4d,h; fig. 5c; fig. 6i-j. We have included the following sentence to **“Quantification and statistical analysis”** in the methods section:

“Significance levels were adjusted for multiple testing using the Bonferroni method for figures 4d, 4h, 5c, 6i-j.”

Reviewer 4:

In the methods, it was indicated that WES was analyzed in hg19, but RNASeq in GRCh38. It may not be critical but was there any particular reason for using two different reference assemblies?

Author response:

WES data was generated before implementation of GRCh38 at our NGS core facility - this has just been implemented for WES recently. The RNA-seq data was generated using GRCh38, which was implemented in our RNA-Seq data analysis pipeline earlier. In our analyses only processed data output, e.g. total number of mutations and gene expression subtype is compared, and the general patterns and correlations we investigate would therefore not be expected to differ with the usage of a GRCh38 version of our WES data. Consequently, we have not reanalyzed the WES data using

GRCh38 as no direct comparisons of variants at specific genomic positions are made between WES and RNA-Seq data.

References

1. Eisenhauer, E. A. *et al.* New response evaluation criteria in solid tumours: revised RECIST guideline (version 1.1). *Eur. J. Cancer* **45**, 228–247 (2009).
2. Petrelli, F. *et al.* Correlation of pathologic complete response with survival after neoadjuvant chemotherapy in bladder cancer treated with cystectomy: a meta-analysis. *Eur. Urol.* **65**, 350–357 (2014).
3. Rosenblatt, R. *et al.* Pathologic downstaging is a surrogate marker for efficacy and increased survival following neoadjuvant chemotherapy and radical cystectomy for muscle-invasive urothelial bladder cancer. *Eur. Urol.* **61**, 1229–1238 (2012).
4. Winkvist, E. *et al.* Neoadjuvant chemotherapy for transitional cell carcinoma of the bladder: a systematic review and meta-analysis. *J. Urol.* **171**, 561–569 (2004).
5. Dash, A. *et al.* A role for neoadjuvant gemcitabine plus cisplatin in muscle-invasive urothelial carcinoma of the bladder: a retrospective experience. *Cancer* **113**, 2471–2477 (2008).
6. Grossman, H. B. *et al.* Neoadjuvant chemotherapy plus cystectomy compared with cystectomy alone for locally advanced bladder cancer. *N. Engl. J. Med.* **349**, 859–866 (2003).
7. Niu, B. *et al.* MSIsensor: microsatellite instability detection using paired tumor-normal sequence data. *Bioinformatics* **30**, 1015–1016 (2014).
8. Middha, S. *et al.* Reliable Pan-Cancer Microsatellite Instability Assessment by Using Targeted Next-Generation Sequencing Data. *JCO Precis Oncol* **2017**, (2017).

REVIEWERS' COMMENTS:

Reviewer #1 (Remarks to the Author):

The authors have sufficiently addressed my concerns.

I note only some small formatting problems in the 2nd table in supplementary figure 1.

Reviewer #2 (Remarks to the Author):

The authors have adequately addressed my previous concerns

Reviewer #4 (Remarks to the Author):

All my previous questions have been appropriately addressed.